# Flavonoids Present in Propolis in the Battle against Photoaging and Psoriasis

**DOI:** 10.3390/antiox10122014

**Published:** 2021-12-19

**Authors:** Claudia Rebeca Rivera-Yañez, Porfirio Alonso Ruiz-Hurtado, María Isabel Mendoza-Ramos, Julia Reyes-Reali, Gina Stella García-Romo, Glustein Pozo-Molina, Aldo Arturo Reséndiz-Albor, Oscar Nieto-Yañez, Adolfo René Méndez-Cruz, Claudia Fabiola Méndez-Catalá, Nelly Rivera-Yañez

**Affiliations:** 1Facultad de Estudios Superiores Iztacala, Universidad Nacional Autónoma de Mexico, Tlalnepantla 54090, Mexico; rbkrivera14@gmail.com; 2Laboratorio de Toxicología de Productos Naturales, Departamento de Farmacia, IPN, Escuela Nacional de Ciencias Biológicas, Av. Wilfrido Massieu, Gustavo A. Madero 07738, Mexico; alonsoruiz55@gmail.com; 3Carrera de Médico Cirujano, Facultad de Estudios Superiores Iztacala, Universidad Nacional Autónoma de Mexico, Tlalnepantla 54090, Mexico; merisam06@iztacala.unam.mx (M.I.M.-R.); reali@unam.mx (J.R.-R.); garciaromogina@gmail.com (G.S.G.-R.); glustein@iztacala.unam.mx (G.P.-M.); oscar.nieto@iztacala.unam.mx (O.N.-Y.); armendez@unam.mx (A.R.M.-C.); 4Laboratorio de Inmunología, Unidad de Morfofisiología y Función, Facultad de Estudios Superiores Iztacala, Universidad Nacional Autónoma de Mexico, Tlalnepantla 54090, Mexico; 5Laboratorio de Genética y Oncología Molecular, Laboratorio 5, Edificio A4, Facultad de Estudios Superiores Iztacala, Universidad Nacional Autónoma de Mexico, Tlalnepantla 54090, Mexico; 6Laboratorio de Inmunidad de Mucosas, Sección de Estudios de Posgrado e Investigación, Escuela Superior de Medicina del Instituto Politécnico Nacional, Salvador Díaz Mirón y Plan de San Luis S/N, Miguel Hidalgo, Casco de Santo Tomas, Mexico City 11340, Mexico; aresendiza@ipn.mx; 7División de Investigación y Posgrado, Facultad de Estudios Superiores Iztacala, Universidad Nacional Autónoma de Mexico, Tlalnepantla 54090, Mexico

**Keywords:** flavonoids, propolis, photoaging, psoriasis, oxidative stress, antioxidant enzymes

## Abstract

The skin is the main external organ. It protects against different types of potentially harmful agents, such as pathogens, or physical factors, such as radiation. Skin disorders are very diverse, and some of them lack adequate and accessible treatment. The photoaging of the skin is a problem of great relevance since it is related to the development of cancer, while psoriasis is a chronic inflammatory disease that causes scaly skin lesions and deterioration of the lifestyle of people affected. These diseases affect the patient’s health and quality of life, so alternatives have been sought that improve the treatment for these diseases. This review focuses on describing the properties and benefits of flavonoids from propolis against these diseases. The information collected shows that the antioxidant and anti-inflammatory properties of flavonoids play a crucial role in the control and regulation of the cellular and biochemical alterations caused by these diseases; moreover, flavones, flavonols, flavanones, flavan-3-ols, and isoflavones contained in different worldwide propolis samples are the types of flavonoids usually evaluated in both diseases. Therefore, the research carried out in the area of dermatology with bioactive compounds of different origins is of great relevance to developing preventive and therapeutic approaches.

## 1. Introduction

The skin is the largest organ of the body and is subjected to oxidative stress daily [1]. Because it is a multifunctional organ, its appearance largely reflects the health and efficacy of its underlying structures [2]. Two main skin layers (the epidermis on the outside and the dermis in the middle) are exposed to most molecules that cause aging; when damage from these molecules reaches the dermis, there is a disruption of the skin’s extracellular matrix resulting in fine wrinkles due to the reduction of collagen, elastic fibers, and hyaluronic acid [3].

The skin protects the body’s internal organs by acting as a barrier against many of the harmful effects of solar ultraviolet (UV) radiation [4,5]. Photoaging is the result of chronic UV radiation exposure from the sun [6]. UV rays can be divided into three segments according to the wavelengths of radiation: short wave (UVC; 200–290 nm), medium wave (UVB; 290–320 nm), and long wave (UVA; 320–400 nm). Each affects the different layers of the skin. Each of these spectra has its own characteristic ability to efficiently penetrate the epidermal and dermal layers of human skin [4].

There are many strategies that the skin uses to combat photoaging caused by different UV rays. However, the primary strategy for prevention of photoaging is photoprotection, and the secondary treatment is by the use of exogenous antioxidants and other compounds that cannot be synthesized in our body [7].

It is important to note that many external agents have the ability to cause skin diseases. Some of these conditions have a diverse etiology, such as the disease known as psoriasis, which is a skin condition characterized by the abnormal proliferation of keratinocytes in the epidermis and the infiltration of different immune cells in the dermis due to an innate and adaptive immune dysfunction [8,9]. Psoriasis mainly affects the skin barrier [10], causing dull red scaly plaques, well defined in the skin, particularly in the extensor prominences and in the area of the scalp [11]. It is one of the most common chronic skin diseases that require long-term treatment [9], affecting 2–3% of the world population [12].

Currently, there are diverse therapies for the treatment of moderate to severe psoriasis that include phototherapy, systemic agents such as methotrexate and cyclosporine, oral treatments such as apremilast, and topical therapies. Nevertheless, although there are many treatments that can be effective and well tolerated, patients affected by psoriasis often do not achieve a clearance of the skin affected by this disease, and, therefore, they do not present symptom relief or improvements in their quality of life. The above have negative effects in patients, so, when receiving ineffective or poorly tolerated treatments for a long time, this can lead to the development of a sustained underlying inflammation, as well as to the deterioration of skin signs and symptoms, and, on the other hand, patients can develop comorbidities related with psoriasis, such as psoriatic arthritis, metabolic syndrome, obesity, diabetes, hypertension, and cardiovascular disease, among others [13]. Therefore, it is vitally important to find new candidates for topical administration in order to reduce side effects and achieve greater therapeutic efficacy [8].

An example of a substance of natural origin with potential for the treatment of skin diseases, caused either by the effect of UV rays or by multifactorial causes such as psoriasis [9], is propolis, which is a natural resinous product that bees make using material obtained from multiple botanical sources. It is mixed with beeswax and enzymes secreted by bees through their salivary glands [14]. Propolis is a sticky, flexible, and soft resin at warm temperatures, or brittle and hard at cold temperatures, it also presents various colors, such as red, green, or brown [15,16]. Generally, the chemical composition of propolis is 50% resin, 30% wax, 10% essential oils, 5% pollen, and 5% other substances [17]. This natural product has been reported to present approximately 300 different compounds [18]. The characteristic chemical groups characterized in propolis are flavonoids, steroids, phenolic acids or their esters, terpenes, stilbenes, aromatic aldehydes, and alcohols as well as fatty acids [18,19]. It is also known that both the chemical composition and the biomedical effect of propolis have a very high variability according to the collection region, the sources of vegetation in the area, and the seasons [20,21]. Moreover, its antioxidant, anti-inflammatory, antibacterial, antidiabetic, and anticancer effects are well known [22,23,24,25,26,27,28,29,30,31,32,33].

Of the different components that make up propolis, flavonoids stand out for having a great anti-inflammatory and antioxidant activity, among others. Furthermore, the plant sources of the flavonoids are highly variable since they depend directly on the geographic region and the flora that bees visit, in fact, the presence and abundance of flavonoids is highly variable in propolis from different countries [20,21]. The substances in propolis with the biological activity mentioned above are known as flavonoids. The flavonoids have a well-known chemical structure which consists of 15 carbon atoms that are arranged to form three aromatic rings named A, B, and C. The B-ring is linked to the A-ring by a three-carbon bridge that binds with one oxygen and two carbons of the A-ring thus forming the C-ring (Figure 1). Based on the different functional groups and level of oxidation in the C-ring, and different connections between B-ring and C-ring, flavonoids are classified into several groups, such as flavones, flavonols, flavan-3-ols, isoflavones, flavanones, anthocyanidins, chalcones, and aurones [34]. These compounds, also known as bioflavonoids, are widely distributed in the plant kingdom [35]. To date, approximately 5000 different types are known [36], with a wide variety of biological properties, among which are anti-photoaging effects on the skin [37,38] and anti-psoriatic activity [35,39].

Therefore, due to the beneficial properties of flavonoids on health, in this review we will focus on describing the most relevant biological activities of the flavonoids present in propolis with regard to the regulation of the cellular, biochemical, and genetic alterations that are generated during the development of photoaging and psoriasis.

## 2. Anti-Photoaging and Photoprotective Properties of Flavonoids

In addition to chronological aging, some extrinsic factors as pollutants, cigarette smoke, and exposure to sunlight can accelerate skin aging [40,41,42]. The epidermal thickening, deep wrinkles, dryness, and loss of elasticity are features of photoaged skin [40,41,42]. Histologically, photodamaged skin presents an accumulation of amorphous elastic fibers and disorganized and fragmented collagen in the dermis [43,44]. Photoaging is the result of chronic UV radiation exposure from the sun [6]. UV rays can be divided in UVC, UVB, and UVA, according to the wavelengths of radiation [4]. UVC is absorbed in the stratosphere, and, therefore, only UVA and UVB reach the surface of the earth (95% UVA and 5% UVB approximately) [6]. UVA irradiation triggers stimulation of matrix metalloproteinase-1 (MMP-1) responsible for collagen degradation, characteristic of photoaged skin [45]. UVB can damage DNA directly through the formation of cyclobutane pyrimidine dimers (CPDs), this action is also observed by UVA rays but to a much lower extent. Furthermore, both UVA and UVB can damage DNA indirectly through the generation of reactive oxygen species (ROS) [46,47]. Different pro-inflammatory cytokines, oxidative stress, including the unbalance of the activity of various antioxidant enzymes, the generation of ROS, and lipid peroxidation are also related to skin damage. A large number of signaling pathways are involved in skin photoaging, such as NF-κB (nuclear factor kappa B), MAPK (mitogen-activated protein kinase), Nrf2 (nuclear factor erythroid 2-related factor 2), STAT (signal transducer and activator of transcription), and JAK (Janus kinase) [48], along with others that will be mentioned later. In an effort to better understand the aging process and cell damage, various cell models have been proposed, such as the human dermal fibroblasts (HDFs), human keratinocytes (HaCaT), and primary human keratinocytes culture (PHKC), to name a few. Similarly, skin photoaging in distinct animal models, such as rats, mice, and even hairless mice, has been investigated. In this regard, the primary strategy for prevention against photoaging is photoprotection, and the secondary treatment is by the use of exogenous antioxidants and other compounds that cannot be synthesized in our body [7], for example, flavonoids. Next, we will address the anti-photoaging properties of various flavonoids on both in vitro and in vivo models.

### 2.1. Flavones

Flavones are a subclass of flavonoids. These compounds present in the C ring a ketone in position 4, and a double-bound between positions 2 and 3 [49]. The effect of different flavonoids such as luteolin, chrysin, and apigenin on UVA-irradiated HDFs, reporting an inhibition on MMP-1 mRNA levels in a dose-dependent manner (1, 5, or 10 µM) was investigated. All flavonoids presented antioxidant activity, and these two properties were correlated with the number of hydroxyl groups in the structure of the compounds. Those with more hydroxyl groups could be used to prevent and/or treat skin aging [50] (Table 1).

In one research, luteolin administration (4 µg/mL) in UVA-irradiated HDFs decreased MMP-1 expression, IL-6 secretion, and hyaluronidase activity through interference with the p38 MAPK pathway. In the same study, luteolin treatment (8 µg/mL) in HaCaT cells exposed to simulated solar radiation (SSR) suppressed IL-20 production, also through interference with the p38 MAPK pathway. In addition, when HaCaT supernatants pretreated with luteolin (8 µg/mL) and exposed to SSR were added to non-irradiated HDFs, MMP-1 and IL-6 expression decreased. Moreover, it was observed that luteolin pretreatment (8 µg/mL) in skin explants exposed to SSR decreased MMP-1 and IL-6 expression, suggesting regulation of these molecules by luteolin, which could mitigate photoaging in the skin [51]. Particularly, the methodology used in this study is very interesting since they use human skin explants, which is a good study model to largely recreate various aspects of photoaging pathology, however, the authors only evaluated MMP-1 and IL-6, when they could have gotten much more information on other parameters. Oliveira et al. propose a model from skin explants obtained from healthy donors who underwent otoplasty surgery, where they performed a histological evaluation and measured ROS, MDA, MMP-1, MMP-8, and MPO, as well as macrophage in the tissue; the evaluation of all these parameters allows a better explanation of the mechanisms regulated by different treatments. They even used levels of UVA exposure compatible with a summer in Brazil, avoiding the models of excessive UVA radiation, which is not compatible with our daily sun exposure [52]. Ex vivo studies allow to recreate the mechanisms of photoaging to a great extent, however, these have their benefits and limitations since no in vitro, in vivo, or ex vivo model is completely faithful to the pathological and clinical conditions of human diseases. To eliminate these limitations, it will be necessary to conduct well-controlled clinical studies.

Luteolin pre-administration (16 μg/mL) in UVB-irradiated HaCaT cells inhibited the CPDs and ROS production and suppressed the p38 MAPK and extracellular signal-regulated kinase (ERK) activation. In addition, this flavonoid decreased the cyclooxygenase-2 (COX-2) expression and the prostaglandin E_2_ (PGE_2_) synthesis, suggesting the potent antioxidant and anti-inflammatory activity of luteolin protects skin from damage occasioned by UV radiation, at least in part by inhibition of the MAPK pathway [53]. Similarly, luteolin administration (5 or 10 μM) in UVB-irradiated HaCaT cells suppressed MMP-1 expression, the activator protein-1 (AP-1), c-Fos activation, the c-Jun phosphorylation, p90RSK, c-Jun N-terminal-kinase 1 (JNK1) activity, and MMP-1 expression, indicating the potent photoaging activity of this flavonoid to prevent or treat skin damage [54].

Other investigations studied the activities of luteolin and apigenin on UVA-irradiated HaCaT cells and observed that treatment with both flavonoids (1–5 mM) suppressed ROS generation and MMP-1 production in these cells. Furthermore, the luteolin and apigenin inhibited some upstream regulators of AP-1, such as c-Jun and c-Fos expression and MAPK phosphorylation. In addition, both flavonoids reduced the influx of Ca^2+^ into HaCaT cells and Ca^2+^/calmodulin-dependent kinase phosphorylation, indicating the effect of luteolin and apigenin in suppressing MMP-1 production by interfering with Ca^2+^-dependent AP-1 and MAPKs signaling and that both flavonoids could be potential candidates for preventing and/or treating skin photoaging [55]. Luteolin is capable of regulating some important pathological mechanisms of photo aging such as the generation of ROS and MMP-1, this is important to continue studying this flavone and to be able to establish it as a complementary treatment in the future, because, these activities are also present in some retinoids, such as vitamin A, E, and safranal, among others; which are currently some of the main candidates in the treatment against photoaging [56,57].

Chrysin administration (6.25, 12.5, or 25 µM) in UVB-irradiated HDFs increased collagen I secretion, reduced glutathione (GSH) level, and decreased the malondialdehyde (MDA) and MMP-1 levels, suggesting an anti-aging effect by regulating the oxidative damage caused by UV radiation, so chrysin could be a promising candidate for protecting against skin damage [38]. A group of researchers informed the effect of chrysin administration (1, 3, or 10 μM) in UVA- and UVB-irradiated HaCaT cells and found that this flavonoid decreased the ROS generation, the COX-2 expression, and the apoptosis. In addition, chrysin increased aquaporin 3 (AQP-3) expression and suppressed p38 and JNK activation, indicating a protective effect of this flavonoid to combat skin damage occasioned by UV radiation, so it could be a promising agent to treat photoaging [58]. Although the effects demonstrated by chrysin are positive and interesting, there is still little information regarding its anti-photoaging effects, so it is necessary to carry out more in vitro and in vivo studies to evaluate the different alterations related to photoaging that justify proposing chrysin as a serious alternative against this condition.

In one study, baicalein administration (10 μmol/L) on UVB-irradiated HDFs inhibited the 12-lipoxygenase and its product, 12-hydroxyeicosatetraenoic acid, and, thus, inactivated transient receptor potential vanilloid 1 (TRPV1) channels and caused a decrease in intracellular Ca^2+^, which, consequently, led to the inhibition of ERK phosphorylation and resulted in the suppression of MMP-1 expression, suggesting an effect of this flavonoid through the TRPV1-Ca-ERK pathway. More studies focused on the activity of baicalein on 12-lipoxygenase could help in the therapy against photoaging of the skin caused by UV radiation [59]. Baicalin treatment (25 μg/mL) in UVB-irradiated HDFs reduced the level of proteins associated with senescence such as p16^INK−4a^, p21^WAF−1^, p53, and γ-H2AX, indicating the activity of this flavonoid against the photoaging caused by UV radiation [60].

The effects that different flavonoids can display in skin photoaging have been tested by some research works in in vivo models. Baicalin topical application (0.5 or 1 mg/cm^2^) in UVB-irradiated mice decreased skin thickening, reduced MMP-3 and MMP-1 expression, and increased collagen III and I production, suggesting the potent activity of this flavonoid against photoaging; thus, it could use to treat skin damage caused by UV radiation [60]. Baicalin pre-administration (50 µg/mL) in UVA-irradiated HDFs increased the telomere length, the transforming growth factor-β1 (TGF-β1) secretion, the glutathione peroxidase (GPx) and superoxide dismutase (SOD) levels, and the c-myc mRNA expression and protein level. Furthermore, this flavonoid decreased MDA levels, p16, p53, p66, tissue inhibitor of metalloproteinase (TIMP-1), MMP-1 mRNA expression, and the p16 and p53 proteins level, suggesting that baicalin has an antioxidant activity and an effective protective effect on the skin against photoaging caused by UV radiation [61]. Baicalin administration (200 μg/mL) in UVB-radiated HaCaT cells suppressed the CPDs and apoptosis. In addition, this flavonoid reduced the c-fos and p53/p21 mRNA expression, p53, repair protein A (RPA), proliferating cell nuclear antigen (PCNA) proteins level, and tumor necrosis factor-α (TNF-α) and IL-6 secretion, indicating the activity of baicalin against skin photoaging caused by UV radiation [62]. It is notable that baicalin has a great antioxidant effect and that it is capable of regulating the levels of oxidative stress and other various parameters in in vitro and in vivo models, which is very favorable in the search for alternatives that complement the therapy against photoaging. Since today the use of sunscreens, protective clothing, avoiding sun’s harmful radiation thereby reducing the progression of skin aging. However, it is precisely molecules with antioxidant properties that help to develop resistance to oxidative stress and slows down the process of skin aging; therefore, baicalein and flavonoids with antioxidant properties are a serious proposition in photoaging therapy [63].

**Table 1 antioxidants-10-02014-t001:** Anti-photoaging properties of flavones reported in in vitro and in vivo models.

Flavones	Model/UV Radiation	Activities	Ref.
Luteolin, chrysin, and apigenin	HDFs/UVA	All flavonoids inhibit the MMP-1 mRNA levels; presents antioxidant activity	[50]
Luteolin		Decreases the MMP-1 expression, the IL-6 secretion, and the hyaluronidase activity	[51]
	HaCaT cells/SSR	Suppresses the IL-20 production; the HaCaT supernatants pretreated and added to fibroblasts decrease MMP-1 and IL-6 expression	
	Skin explants/SSR	Decreases MMP-1 and IL-6 expression	
	HaCaT cells/UVB	Inhibits CPDs and ROS production; suppresses the p38 MAPK and ERK activation; decreases COX-2 expression and the PGE_2_ synthesis	[53]
		Suppresses MMP-1 expression, AP-1 and c-Fos activation, c-Jun phosphorylation, p90RSK and JNK1 activity, and MMP-1 expression	[54]
Luteolin and apigenin	HaCaT cells/UVA	Both suppress ROS generation and MMP-1 production; inhibit c-Jun and c-Fos expression and the MAP kinases phosphorylation; reduce the influx of Ca^2+^ and Ca^2+^/calmodulin-dependent kinases phosphorylation	[55]
Chrysin	HDFs/UVB	Increases collagen I secretion and the GSH level; decreases the MDA and MMP-1 level	[38]
	HaCaT cells/UVA and UVB	Decreases ROS generation, COX-2 expression, and apoptosis; increases AQP-3 expression; suppresses p38 and JNK activation	[58]
Baicalein	HDFs/UVB	Inhibits the 12-lipoxygenase and through the TRPV1-Ca-ERK pathway suppresses the MMP-1 expression	[59]
Baicalin		Reduces the p16^INK−4a^, p21^WAF−1^, p53, and γ-H2AX proteins level	[60]
	C57BL/6 mice/UVB	Decreases skin thickening; reduces MMP-3 and MMP-1 expression; increases collagen III and I production	
	HDFs/UVA	Increases the telomere length, the TGF-β1 secretion, the GPx and SOD levels, and the c-myc mRNA expression and protein level; decreases MDA levels, p16, p53, p66, TIMP-1, MMP-1 mRNA expression, and the p16 and p53 proteins level	[61]
	HaCaT cells/UVB	Suppresses the CPDs and apoptosis; reduces the c-fos and p53/p21 mRNA expression, the p53, the RPA, and PCNA proteins level, and TNF-α and IL-6 secretion	[62]
	BALB/c mice/UVB	Reduces skin hyperplasia and edema, hydrogen peroxide generation, and photolesions formation	[64]
	C3H/HeN mice/UVA	Decreases the CD11b^+^Gr1^+^ myeloid-derived suppressor cells, the TLR4 expression level, TRAF6, IRAK4, and MyD88 protein expression, TIRAP and MyD88 mRNA expression, MMP-1 and MMP-9 expression levels, COX-2, IL-1β, IL-10, and iNOS	[65]

The evaluation of the topical pre- or post-treatment activities of baicalin (1 mg/cm^2^ skin area) on light UVB-irradiated mice skin showed a reduction in skin hyperplasia and edema, hydrogen peroxide generation, and photolesion formation, suggesting that this flavonoid is a candidate in the treatment of damage caused by UV exposure [64]. Similarly, baicalin pre- and post-treatment (4 mg) on UVA-irradiated mice decreased the CD11b^+^Gr1^+^ myeloid-derived suppressor cells and the TLR4 expression level on CD11b^+^. Furthermore, this flavonoid decreased the protein expression of TNF receptor associated factor 6 (TRAF6), interleukin 1 receptor associated kinase 4 (IRAK4), and myeloid differentiation primary response 88 (MyD88), and the mRNA expression of Toll-interleukin 1 receptor (TIR) domain-containing adapter protein (TIRAP) and MyD88 involved in the TLR4 pathway. In addition, baicalin decreased inflammatory biomarkers such as COX-2, IL-1β, IL-10, and inducible nitric oxide synthase (iNOS), and MMP-1 and MMP-9 expression levels, suggesting the activity of this flavonoid to protect mice skin against inflammation and oxidative damage, probably through the TLR4 pathway, so baicalin could be used to protect skin from UV light damage [65]. It can be see that the administration of this flavonoid acts on cytokines and transcription factors that are directly related to alterations caused by exposure to UVA and UVB rays, but the most relevant aspect is that they suggest that baicalein regulates these alterations by signaling the TLR4, which opens the door to carry out studies in combination with other compounds such as quercetin, resveratrol, apigenin and luteolin [66,67,68,69] in order to find a synergistic effect for a better regulation of this signaling pathway in photoaging.

### 2.2. Flavonols

In simple terms, flavonols are flavonoids that have a ketone group; in comparison with the flavones, this subgroup presents in the C ring a hydroxyl in position 3 [49]. Various studies have reported the effect of different flavonols, such as quercetin, myricetin, and kaempferol, on UVA-irradiated HDFs, reporting an inhibition on MMP-1 mRNA levels in a dose-dependent manner (1, 5, or 10 µM). All flavonols presented antioxidant activity, and these two properties were correlated with the number of hydroxyl groups in the structure of the compounds. Those with more hydroxyl groups could be used to prevent and/or treat skin aging [50] (Table 2). The activity of galangin, kaempferol, and quercetin was demonstrated on reconstituted human skin tissue (EpiDerm^TM^). Quercetin treatment (100 μM or 200 μM) in UVA- or UVB-irradiated EpiDerm^TM^ reduced MMP-1 and TNF-α secretion. Moreover, pretreatment with the three flavonoids (52 μM) reduced the cyclobutene thymine dimers (CTDs) in UVB-irradiated EpiDerm^TM^, suggesting a protective effect of the galangin, kaempferol, and quercetin against UV-radiation skin damage through its antioxidant activity [70]. Myricetin pre-administration (3, 10, and 30 µM) in UVB-irradiated PHKC decreased the MDA level and suppressed H_2_O_2_ production and JNK activation, suggesting protection by this flavonoid from the damage caused by UV radiation, so myricetin could be a potential candidate for preventing or treating skin photoaging [71]. The effect of the topical administration of myricetin (1 or 5 nmol) conjugated beads in UVB-irradiated hairless mice was described, a decrease in epidermal thickening and an inhibition of enzyme activity and MMP-9 protein expression was found. Also, this flavonoid inhibited the Raf (rapidly accelerated fibrosarcoma) kinase activity and consequent decrease in MEK and ERK phosphorylation in mouse skin, indicating a potential anti-photoaging effect, modulating MMP-9 expression by the inhibition of Raf kinase activity [72]. The results reported for galangin and kaempferol are good and favorable, however they are still very partial and not very integrative to be considered a real alternative against photoaging. Therefore, it is still necessary to carry out more studies focused on evaluating more characteristics of this disease, such as the levels of ROS, MPO, pro-inflammatory cytokines and even the use of models that allow a histological analysis of the tissue [73]. The challenges for galangin, kaempferol, and myricetin are still several, but the future investigations will help conclude if these flavonoids can be used against photoaging.

Similarly, quercetin pre-administration (10 μg/mL) in UVB-irradiated HaCaT cells decreased NF-κB protein in the nucleus, and quercetin nanoparticles pre-administration (80 μg/mL) decreased this protein more than quercetin alone. Also, both pre-administrations suppressed NF-κB inhibitor-α (IkB-α) phosphorylation and were more effective than quercetin nanoparticles; consequently, the COX-2 protein expression level was reduced. In addition, in this same research, it was observed that quercetin nanoparticles pre-administration (4 mg/mL) in light UVB-irradiated mice was more effective than quercetin alone pre-administration (0.5 mg/mL) in decreasing IkB-α phosphorylation, COX-2 expression, and PGE_2_ concentration in the mice skin. Taking these activities into account, it could be suggested that both treatments suppress the NF-κB/COX-2 signaling pathway; therefore, they could be proposed as candidates for therapy against skin damage caused by UV radiation [74].

Quercetin administration (20 μg/mL) in UVA- and UVB-irradiated PHKC reduced the NF-kB DNA-binding and, consequently, inhibited TNF-α, IL-1β, IL-6, and IL-8 expression, indicating an anti-inflammatory activity by this flavonoid in mice skin to combat the damage caused by UV radiation, probably due to the inhibition of the NF-κB pathway [75]. Similarly, administration of pearls containing quercetin (5–40 μM) on light UVA- and UVB-irradiated human abdominal skin tissue inhibited COX-2, MMP-1, and collagen degradation. Also, this flavonoid suppressed NF-κB and AP-1 activation, and decreased protein kinase B (Akt), JNK, ERK, and STAT3 phosphorylation. In addition, the quercetin suppressed the JAK2 and protein kinase C δ (PKCδ) activity. Therefore, quercetin exhibits activity that protects the skin from inflammation and photoaging and could be a potent candidate to treat skin damage caused by UV radiation [37]. The effect of the administration of a topical emulsion containing 1% of quercetin to the dorsal skin of light UVB-irradiated hairless mice has been studied, and a decrease of myeloperoxidase (MPO) activity, an increase of GSH, and suppression of proteinases secretion/activity were observed, which suggests that the topical application of this flavonoid could be used to reduce skin damage caused by UVB radiation [76].

Quercitrin pre-administration (10, 20, or 40 μM) in UVB-irradiated mouse epidermal (JB6) cells suppressed the apoptosis, reduced the cleaved caspase-3 (C-caspase-3), and cleaved PARP1 (C-PARP1) activation. In addition, this flavonoid decreased the NF-κB activation in nuclei, DNA damage in cells, 8-hydroxy-2′-deoxyguanosine (8-OHdG) production, γ-H2AX expression, and superoxide radical production. Furthermore, the quercitrin inhibited hydroxyl radical and hydrogen peroxide production and increased SOD and catalase (CAT) expressions. The authors of this investigation found that quercitrin topical pre-treatment (1 mg) in UVB-irradiated hairless mice decreased apoptosis in mouse skin, C-caspase-3 and C-PARP1 expressions, DNA damage in mice skin epidermal cells, 8-OHdG production, and γ-H2AX expression. In addition, this flavonoid increased XPA (DNA repair gene) expression in mice skin, SOD and CAT expressions, and GSH levels, suggesting the potential anti-inflammatory and antioxidant activity of quercitrin; therefore, it could protect the skin from oxidative damage caused by UV radiation [77]. Quercetin is a very abundant flavonol in plant sources, making it one of the most studied flavonoids and for which there is sufficient evidence in vitro and in vivo studies of its beneficial effects on health. The effects of quercetin and its derivatives on photoaging should be considered as a viable and interesting alternative since they have effects at different levels of the pathological processes of photoaging. However, one of its main limitations is to find a vehicle or administration route that ensures its correct absorption since it is very limited. In this sense, various strategies have been developed, of which the design of nanoparticles with quercetin stands out, which apparently improve their assimilation [78,79,80]. Therefore, it is necessary to carry out more studies that can demonstrate whether nanoparticles with this flavonol improve the effects on ROS, activation of signaling pathways (NF-κB, Akt, JNK, ERK, and AP-1), and the production of pro-inflammatory cytokines. At the same time, it is urgent that well-controlled clinical studies be carried out in order to find an adequate dose that guarantees the desired pharmacological effects without adverse effects.

Fisetin treatment (5−25 μM) in UVB-irradiated HDFs inhibited collagen degradation, MMP-9, MMP-3 and MMP-1 expression, COX-2 and nitric oxide (NO) generation, PGE_2_, and intracellular ROS. Also, this flavonoid decreased JNK and ERK expression, p38 phosphorylation, and IκB degradation. Moreover, fisetin inhibited NF-κB translocation into the nucleus and the cAMP-response element binding protein (CREB) phosphorylation level in the phosphatidylinositol-3-kinase (PI3K)/Akt/CREB pathway, indicating the photoaging activity of this flavonoid, which could be proposed as a candidate to protect skin from damage caused by UV radiation [81].

### 2.3. Flavanones

The group of the flavanones have the C ring saturated and also are named dihydroflavones because the only difference with the flavones is that the double bond between positions 2 and 3 is saturated [49]. In 2021, Lohakul et al. [82] reported that hesperetin pre-administration (2.5, 5, or 10 μg/mL) in UVA-irradiated HDFs decreased ROS generation and also reduced collagen depletion and MMP-1 activity. Additionally, this flavonoid positively regulated Nrf2 activity and its genes NAD(P)H: quinone oxidoreductase 1 (NQO-1) and glutathione S-transferase (GST) (Table 3). In addition, in an in vivo model of this same research, they found that hesperetin topical pretreatment (0.3, 1, or 3 mg/cm^2^) in light UVA-irradiated mice skin decreased collagen loss and MMP-1 activity. Furthermore, this flavonoid increased nuclear Nrf2 levels and its NQO-1 and GST target proteins. Additionally, the hesperetin decreased the 8-OHdG, an indicator of oxidative DNA damage, in mice skin, indicating an attenuation of photoaging through the ability of hesperetin to positively regulate Nrf2; therefore, this flavonoid could be a promising agent to restore redox equilibrium and help in treatment against skin photoaging [82].

**Table 3 antioxidants-10-02014-t003:** Anti-photoaging activity of the flavanones.

Flavanones	Model/UV Radiation	Activities	Ref.
Hesperetin	HDFs/UVA	Decreases ROS generation; reduces collagen depletion and MMP-1 activity; positively regulates Nrf2 activity and its genes NQO-1 and GST	[82]
	BALB/c mice/UVA	Decreases collagen loss, MMP-1 activity, and 8-OhdG; increases nuclear Nrf2 levels and its NQO-1 and GST target proteins	
Hesperidin	HaCaT cells/UVA	Increases SOD activity and total antioxidative capacity levels; reduces MDA content; decreases IL-1β, IL-6, and TNF-α mRNA levels and proteins expression	[83]
	Hairless mice/UVB	Decreases collagen fiber loss, wrinkle formation, and TEWL; suppresses MMP-9 activity and mRNA levels, TNF-α and IL-8 production, and ERK and MEK phosphorylation	[84]
Naringenin	HaCaT cells/UVB	Suppresses MMP-1 expression, AP-1 activity, FRA1 expression and phosphorylation, p90RSK phosphorylation, and ERK2 activity	[85]
	Hairless mice/UVB	Suppresses TEWL, wrinkle formation, and MMP-13 expression	
		Inhibits the IL-1β, IL-10, IL-6, and TNF-α production; suppresses lipid hydroperoxides and superoxide anion production; preserves GSH levels and CAT activity, Nrf2 mRNA expression, glutathione reductase, and GPx	[86]

Similarly, hesperidin administration (220 μg/mL) in UVA-irradiated HaCaT cells increased SOD activity and total antioxidative capacity levels, reducing MDA content in these cells. Furthermore, this flavonoid decreased IL-1β, IL-6, and TNF-α mRNA levels and proteins expression, indicating the protective effect of hesperidin from skin damage caused by UV radiation in HaCaT cells through mitigation inflammation and oxidative stress in these cells, and, thus, could be proposed as a candidate for sun protection [83]. Hesperidin oral administration (100 mg/kg) in UVB-irradiated hairless mice decreased collagen fiber loss, wrinkle formation, and transepidermal water loss (TEWL). Furthermore, this flavonoid suppressed MMP-9 activity and mRNA levels, TNF-α and IL-8 production, and ERK and MEK phosphorylation, indicating a potential effect against photoaging by suppressing the activity of MMP-9 through the inhibition of MAPK-dependent signaling pathways, which could help protect the skin from damage caused by UV radiation [84]. Hesperidin and hesperetin have similar effects since they decrease ROS levels, collagen degradation and MMP activity, however, in these studies no histological or ex vivo models are used, which would help to have more certainty about the benefits of these flavanones in photoaging. On the other hand, it would be interesting to determine if the co-administration of hesperidin and hesperetin increases their activity and thus achieve better protection against damage caused by UV radiation.

Naringenin pre-administration (5 or 10 μM) in UVB-irradiated HaCaT cells suppressed MMP-1 expression, AP-1 activity, Fos-related antigen 1 (FRA1) expression and phosphorylation, p90RSK phosphorylation, and ERK2 activity, indicating a potential anti-photoaging activity of this flavonoid by inhibiting the ERK2 signaling pathway [85]. In addition, naringenin administration (5 or 10 µM) in UVB-irradiated hairless mice suppressed TEWL, wrinkle formation, and MMP-13 expression, which showed its anti-photoaging activities [85]. The topical administration of a formulation containing naringenin (500 mg) in UVB-irradiated hairless mice demonstrated inhibition of IL-1β, IL-10, IL-6, and TNF-α production. Moreover, the naringenin suppressed lipid hydroperoxides and superoxide anion production, preserved GSH levels and CAT activity. Also, this flavonoid preserved mRNA expression of transcription factor Nrf2, glutathione reductase, and GPx, indicating the potential antioxidant and anti-inflammatory effect of naringenin, which could protect the skin of mice from damage caused by UV radiation [86].

### 2.4. Flavan-3-ols and Isoflavones

The flavan-3-ols are also named dihydroflavonols, due always have in the position 3 of the C ring a hydroxyl group [49]. On the other hand, isoflavones also knowing as phytoestrogens [87]. One study found that epigallocatechin-3-gallate (EGCG) nanoparticles treatment (10 μg/mL) in UVB-irradiated HaCaT cells was more effective than EGCG alone treatment (10 μg/mL) in decreasing intracellular ROS levels, MDA level, and MMP-2 and MMP-9 expression, indicating the potent anti-aging and antioxidant activity of this flavonoid; therefore, it could be proposed as a probable topical therapeutic agent to protect against skin damage caused by ultraviolet radiation [88] (Table 4). EGCG administration (200 μg/mL) in UVB-irradiated HaCaT cells reduced apoptosis and suppressed c-fos, p21, and p53 mRNA expression. Furthermore, this flavonoid decreased TNF-α and IL-6 secretion, indicating anti-inflammatory and anti-photoaging activity; therefore, it could protect from damage caused by UV radiation [89].

In another study, EGCG administration (1, 10, or 20 µM) in UVB-irradiated HDFs inhibited collagen degradation. Pre-administration of this flavonoid also suppressed MMP-13, MMP-8, and MMP-1 production in a dose-dependent manner. Furthermore, EGCG inhibited MAPK, ERK1/2, p38 MAPK, and JNK phosphorylation and apoptosis signal-regulating kinase-1 (ASK-1) activation, demonstrating the capacity of this flavonoid to inhibit the production of different collagenase through the suppression of pathways involved with MAPK, and thus could be a promising candidate to prevent and/or treat skin photoaging [90]. The topical application of a cream that contained 2% EGCG on the skin of rats exposed to UVA radiation showed a diminution in sunburn cells and dermo-epidermal activation, suggesting the protective activity of this flavonoid against UV exposure [91]. The properties of EGCG, like the various flavonoids mentioned in this work, have properties that attend to some of the molecular, biochemical, and cellular disorders caused by photoaging. Which is very interesting since several of the treatments developed with the damage generated by UV radiation have a cosmetic approach instead of a therapeutic one; that is, they are designed to reduce wrinkles and spots on the skin, leaving aside the severe damage at cellular level caused by radiation and ROS, which can cause other diseases, such as skin cancer [92,93,94,95]. Therefore, flavonoids could be proposed in the design of new cosmeceutical products, which have a cosmetic approach as well as a therapeutic one; that is, they attend to health and beauty. These products are relatively new to the dermatology industry so there are few alternatives available on the market.

**Table 4 antioxidants-10-02014-t004:** Main anti-photoaging functions of flavan-3-ols and isoflavones.

Flavonoids	Model/UV Radiation	Activities	Ref.
EGCG and EGCG nanoparticles	HaCaT cells/UVB	Both decrease the intracellular ROS levels, MDA level, and MMP-2 and MMP-9 expression	[88]
EGCG		Reduces apoptosis; suppresses c-fos, p21, and p53 mRNA expression; decreases TNF-α and IL-6 secretion	[89]
	HDFs/UVB	Inhibits collagen degradation, MAPK, ERK1/2, p38 MAPK, and JNK phosphorylation and ASK-1 activation; the pre-administration suppresses MMP-13, MMP-8, and MMP-1 production	[90]
	Rats/UVA	Diminution in sunburn cells and dermo-epidermal activation	[91]
Catechin		Increases CAT, GPx, and SOD levels; decreases TBRAS levels	[96]
Genistein	HDFs/UVB	Reduces the apoptosis and MDA level; increases SOD activity; decreases the p66Shc and FKHRL1 level and phosphorylation	[97]

Treatment with a topical nanogel containing catechin (5 mg) in UVA-irradiated rats increased the levels of skin antioxidant enzymes CAT, GPx, and SOD and decreased thiobarbituric acid reactive substances (TBARS) levels, demonstrating the potent photoprotective effect of this flavonoid; therefore, it could be proposed to protect the skin from oxidative damage caused by UV radiation [96]. Genistein administration (40 or 80 mg/mL) in UVB-irradiated HDFs reduced the apoptosis and MDA level and increased SOD activity. In addition, this isoflavone decreased the p66Shc and FKHRL1 level and phosphorylation, suggesting the protective effect of genistein against damage caused by UV radiation, at least in part due to its antioxidant activity and the regulation of oxidative stress through modulation of the p66Shc-dependent signaling pathway, so genistein could be a potent agent to prevent or treat skin photoaging [97].

The research about flavonoids on the different models mentioned above demonstrates various effects focused on combating skin photoaging, such as the regulation of the activity of different metalloproteinases, pro-inflammatory cytokines, and antioxidant enzymes, as well as ROS generation, lipid peroxidation, collagen degradation or secretion, gene expression or the level of proteins associated with senescence and the modulation of some signaling pathways involved with skin damage. In recent years, studies focused on investigating intestinal absorption and the bioavailability of different flavonoids, such as apigenin, have been carried out to observe their impact on cells irradiated with ultraviolet light [98]. In this regard, it is known that some types of flavonoids administered or applied directly do not show good biological activity, however, an effort has been made to improve the biological activity of flavonoids by investigating different forms of conjugation or combination with different substances, such as sepharose beads, nanoparticles, or gels based on microemulsions, sodium alginate, or poly (vinyl) alcohol, or even liposomes as amphipathic transporters of bioactive compounds, such as, for example, EGCG, quercetin, and naringenin, among others [74,88,99,100,101]. An example of this is the research conducted by Esposito et al. in 2020 [99], who have studied a novel series of hybrid hydrogels at different ratios based on sodium alginate and poly(vinyl) alcohol quercetin-loaded. The permeability of quercetin through the skin showed different penetration/permeation profiles according to the hydrogel’s blend, which will allow the selection of hydrogels ratio for a best local and prolonged skin effect, making the use of these hydrogels a promising solution for the delivery of flavonoids for the treatment of skin ageing and inflammation. Another example is the study carried out in 2020 by Parashar et al. [100], who mention that the topical delivery system for sericin gel loaded with microemulsion containing naringenin, displayed higher retention and deposition of naringenin in the skin. Moreover, this is a propitious topical delivery system for naringenin for preventing or inhibiting UV-induced skin aging, that displayed enhanced therapeutic potential when compared with plain naringenin. Other research mentions that liposomes are nanocarriers that are used to incorporate bioactive compounds or drugs to treat some skin diseases. Liposomes are membranes of different sizes (from micro to nanometers) mainly composed of cholesterol and phospholipids, forming structures similar to cell membranes. Also, they are composed of a lipid bilayer and an aqueous nucleus that gives them their amphiphilic property. Furthermore, to improve permeability in the skin, edge activators are added to the liposomes, which reduces the stiffness of the bilayer structure making it deformable. However, more research is needed regarding the efficacy of loaded liposomes to transport different bioactive compounds to study their release through the skin [101]. It should be noted that it is still a challenge to find and standardize the best method of administration or application of flavonoids in a safe and efficient way to achieve their best expected biological activity for clinical application.

Even though the information is scarce, the potential of the anti-photoaging effect that the administration of the combination of a flavonoid (rutin) with ascorbic acid may have in UVA- and UVB-irradiated fibroblasts and keratinocytes in humans has been studied. Findings suggest that this combination is more efficient in increasing the activity of SOD and CAT, decreasing the formation of ROS, caspases 9, 8, and 3, and NF-κB expression than each of the compounds separately; therefore, this combination could present a synergism and potentiating effect of its different properties [102]. However, the in vitro and in vivo studies in this section do not mention a possible combination of drugs with flavonoids, which leaves in doubt whether there could be a synergistic activity between them to propose a more effective treatment against photoaging. It should be noted that there are also few clinical reports on the anti-photoaging activity of flavonoids, and more research should be carried out, such as that of Choi et al. [103], who found that the application of a cream containing rutin on the skin of patients reduced the number, area, and length of wrinkles and increased the elasticity of the skin.

Therefore, once the multiple potential anti-photoaging properties of all these different flavonoids have been studied in vitro and in vivo, it is essential to conduct more clinical investigations related to the study of new, better, and more efficient therapies that help to counteract skin damage generated by exposure to ultraviolet rays in patients.

### 2.5. Propolis: Bee Product Rich in Flavonoids with Anti-Photoaging Activity

The product that bees make using material obtained from multiple botanical sources known as propolis has a complex chemical composition and is abundant in flavonoids [27,30,104]. In addition, it presents an immense variety of biological properties, among them its anti-photoaging activity [105,106,107,108], which we will review below.

Brazilian green propolis administration (3, 10, or 30 μg/mL) in UVA-irradiated HDFs inhibited intracellular ROS generation and ERK and p38 phosphorylation level, indicating the promising effect of propolis as a possible agent to treat skin damage caused by UV radiation [109] (Table 5). Similarly, Brazilian green propolis pre-administration (30 μg/mL) in UVA-irradiated HDFs positively modulated early Heme oxygenase-1 (HO-1) expression and induced the rapid translocation of Nrf2 to the nucleus, suggesting the anti-photoaging activity of propolis, which could help in the treatment against skin damage caused by UV radiation [105]. One study observed that Iranian propolis pre-administration (100 μg/mL) in UVB-irradiated HDFs raised the nerve growth factor (NGF) and Forkhead box O3A (FOXO3A) gene expression, reduced b-galactosidase activity, and presented outstanding antioxidant activity, indicating the potent effect of this propolis to protect the skin from photoaging, probably through its antioxidant activity and the regulation of the two mentioned genes [110]. Kim et al. [106], in 2020, found that Korean propolis administration (5, 10, or 20 μg/mL) in UVB-irradiated HDFs inhibited MMP-1 production, mRNA levels, and collagen degradation. Furthermore, the propolis suppressed Akt, phosphoinositide-dependent protein kinase-1 (PDK1), and PI3K activity. In addition, several flavonoids were identified in Korean propolis, such as catechin, naringenin, apigenin, and quercetin; these last three compounds (5 μM) inhibited PI3K activity, demonstrating an activity of Korean propolis against skin photoaging caused by UV radiation, at least in part by suppressing PI3K activity. In other study, Korean propolis treatment (5 or 10 μg/mL) in UVA-irradiated HaCaT cells suppressed apoptosis, C-caspase-3 expression, decreased the loss of mitochondrial membrane potential and ROS production, suggesting the protective activity of propolis on skin damage caused by UV radiation, at least in part by its antioxidant and antiapoptotic activities [111]. Greek propolis pre-administration (20 μg/mL) in UVB-irradiated HaCaT cells presented outstanding antioxidant activity and reduced DNA damage and total protein carbonyl content. In UVB-irradiated EpiDerm^TM^, propolis pre-administration (20 μg/mL) decreased MMP-9, MMP-7, MMP-3, and MMP-1 mRNA levels, suggesting the protection of Greek propolis on skin damage caused by UV radiation, probably through its antioxidant and anti-photoaging activities; thus, it could be proposed as a potential candidate for new complementary therapies to treat skin damage [107]. Romanian propolis pre- or post-topical application (3 or 1.5 mg/cm^2^) in UVB-irradiated mice reduced MDA and IL-6 levels, C-caspase-3 activation, sunburn cell formation and CPDs generation, and increased GPx activity, indicating the potent anti-photoaging activity of propolis; therefore, it could be proposed as an agent to protect skin from damage occasioned UV radiation [108].

It should be noted that there is little information about the anti-photoaging activity of propolis; however, the release, penetration, and retention capacity of nanoemulgels prepared with propolis has been investigated in animal models of skin damage by ultraviolet radiation, highlighting the potential of its topical application compared to propolis alone [112,113]. Besides, it is important to highlight that there is very little information regarding the safety of propolis, since it is a subject of study little investigated, it was found that propolis is considered safe by Diniz et al. [114], based on the results obtained in the biochemical safety profile in the research. However, it is clear that more studies are needed on this subject, as well as on many of the natural products used as complementary treatments in various diseases. In addition, it is important to mention that they should start with the implementation of more investigations to establish and propose methods of analysis and extraction of the different compounds present propolis around the world, such as flavonoids, to be able to compare in a more efficient way the biomedical activities of the propolis and its bioactive components [23]. Furthermore, the little research on the safety and standardization of propolis are some of the limitations for its use in clinical practice. It is also indispensable to note that, so far, there are no clinical studies of propolis anti-photoaging activity, and being able to test this product, rich in flavonoids, begins a new field of study focused on investigating the properties of propolis on skin damage caused by ultraviolet radiation and observe its potential in possible treatment in patients. In addition, it is important to point out that in most studies in this section, the chemical composition of the different propolis investigated is not mentioned, and since they are known to be rich in flavonoids, it would be enriching to report the compounds that make up propolis in order to complement these studies with the investigation of its identified flavonoids and to prove which could have the activity to protect the skin from damage caused by ultraviolet rays and even have a synergistic effect with the combination of different drugs.

## 3. Properties of Flavonoids against Psoriasis

Psoriasis is a chronic inflammatory disease considered a condition primarily of the skin, affecting patients physically, mentally, and socially. It is a complex disease with a multifactorial etiology. The interplay between genetics and exposure to a triggering factor(s) results in manifestations of the disease, such as red, scaly, and raised patches on the skin that can present during the lifetime of the patient; remissions and relapses are unpredictable. Psoriasis is characterized by the presence of epidermal hyperproliferation, abnormal keratinocyte differentiation, angiogenesis with blood vessel dilatation, and excess of Th1 and Th17 inflammation in the skin of the affected area. Psoriasis affects 2–3% of the global population [8,39,115,116,117].

Currently, it is known that psoriasis is the result of a disturbance among innate and adaptative cutaneous immune responses. It has been shown that this autoimmune pathology is displayed on an inflammatory background in which the psoriatic plaque can develop in the dermal layer of the skin; therefore, there is an interaction among keratinocytes (which are the cell type that shapes the layer of the skin) with different cells involved in the innate and adaptative immune response. LL37, β-defensins, and S100 proteins are the most studied psoriasis-associated AMPs (antimicrobial peptides), of which, LL37 also named cathelicidin, has been associated with a pathogenic role in psoriasis; this stimulates TLR-9 in plasmacytoid dendritic cells (pDCs); and starting the development of the psoriatic plaque in which there is a stimulation to produce of type I IFN (IFN-α and IFN-β), which, in turn, promotes the phenotypic maturation of myeloid dendritic cells (mDCs), that are implicated in Th1 and Th17 differentiation and function, as well as the production of IFN-γ and IL-17; on the other hand, when activated mDCs migrate into draining lymph nodes, it secrets TNF-α, IL-23, and IL-12; in the case of IL-23 and IL-12 it is knowing that can modulate the differentiation and proliferation of Th17 and Th1, respectively. The maintenance phase of psoriatic inflammation is driven by different T cell subsets in which Th17 cytokines (IL-17, IL-21, and IL-22) activate the proliferation of keratinocytes in the epidermis; moreover, the proliferation of these cells also are stimulated by TNF-α, IL-17, and IFN-γ; and, therefore, keratinocytes are directly implicated in the inflammatory cascade by means of the secretion of different cytokines like IL-1, IL-6, and TNF-α, chemokines, and AMPs [12,118].

Different experimental models to study psoriasis and the related mechanisms to this autoimmune disease have been developed both in vitro and in vivo; these are used by many researchers to investigate the pathogenic mechanisms of psoriasis. Although it is worth mentioning that both experiment models have limitations, such as the absence of blood vessels and microenvironment on in vitro models, and the differences of thickness, epidermis architecture, and immunological mechanisms involved on murine in vivo models with respect to the pathogenesis present in the human dermis. Nevertheless, both types of experimental models have been helpful to test the effectiveness and the development of new therapeutic agents [119].

Flavonoids are polyphenols that are emerging as highly effective, viable alternatives to conventional medicines in the treatment of different diseases, such as psoriasis, because of the psoriasis-related activity that some flavonoids present [8,35,115].

### 3.1. Flavones

Weng et al. [120] reported that luteolin inhibited the effect of IL-8 and IL-6 release in normal human epidermal keratinocytes (NHEKs) and HaCaT cells when both cell lines are stimulated with TNF-α. Interestingly, inhibition produced by luteolin (10 μM) was stronger in NHEKs than in HaCaT cells. Also, both cell lines stimulate the expression of genes that encode the two different NF-κB subunits, encoding NF-κB p50 subunit (NFKB1) and NF-κB p65 subunit, also called encoding v-rel reticuloendotheliosis viral oncogene homolog A (RELA). Therefore, treatment with luteolin of both keratinocytes cell lines decreases mRNA expression of both subunits of NF-κB. Additionally, luteolin reduces the HaCaT cells proliferation, although cell viability remains above 90% and the metabolic activity of cells was not affected by luteolin. Because the decrease of intracellular ATP content was slight, this was not significant with respect to control cells [120] (Table 6). In 2020, Lv et al. [121] found that in HaCaT cells, heat shock protein 90 (HSP90)-induced with interferon-γ (IFN-γ), the luteolin (50 μM) inhibits the transcriptional expression of both HSP90β and HSP90α. Additionally, IFN-γ can promote the exosome secretion of HSP90 in HaCaT cells, while luteolin can reverse this IFN-γ-induced effect. Generally, in vivo evaluation of anti-psoriatic properties of different flavonoids includes the use of the imiquimod-induced psoriatic lesions murine model. The in vivo effect of luteolin shows that the intraperitoneal injection of this flavone (50 mg/kg) in male BALB/c mice, previously psoriasis-induced with imiquimod cream, reduced the psoriatic area and severity index (PASI), histological severity, and HSP90β and HSP90α expression in the psoriatic skin-lesions of experimental animals. This effect was also closely related to the capacity of luteolin to decrease the proportion of T cells (Th17/T regulatory (Treg) and Th1/Th2) in mice treated with this flavonoid, and, therefore, alleviate psoriasis in vivo [121]. In addition, Zhou et al. in 2020 [39], reported that luteolin (from 6.25–200 μM) did not affect the cell viability of Raw264.7 macrophages, and this flavonoid (25, 100, or 200 μM) also decreased both mRNA and protein levels of TNF-α, IL-1β, IL-6, IL-23, and IL-17A pro-inflammatory cytokines in this macrophage cell line stimulated with LPS in a concentration-dependent form. In this context, Raw264.7 macrophages treated with luteolin inhibited NF-κB p65 and COX-2 expression. Additionally, iNOS and NO also were suppressed by the treatment with luteolin. The same study also evaluated luteolin’s topical effects in this induced-psoriasis mice model. The workgroup reported that topical treatment with luteolin (80 mg/kg) showed similar skin morphology to mice treated with the drug tacrolimus, which had smoother skin, reduced skin thickening, superficial erythema, and fewer scales. The PASI measurement was also reduced with respect to the psoriasis model group; in this work, luteolin decreased infiltrated neutrophils (GR1^+^), T cells (CD8^+^), and macrophages (f4/80^+^) in the skin of psoriatic mice. Moreover, proinflammatory cytokine production of skin and eyeball blood was evaluated. Luteolin reduced IL-1β, IL-17A, IL-23, and IL-6 which could contribute to the anti-inflammatory effects of this flavonoid and the reduction of psoriasis damage in the mice model [39].

Palombo et al. [122] studied the isomer luteolin-7-glucoside, which is the most stable form of luteolin found in nature (plants, fruits, and vegetables), to treat human normal keratinocytes (HEKn). This flavonoid altered the cell cycle progression; specifically, cells were accumulated in the G1 phase in a time-dependent and dose-dependent manner. However, the flavonoid did not induce apoptosis in the cells; this flavonoid increased keratin 10 (KRT10) expression (which together with KRT1 provides mechanical stress resistance and reinforces cell–cell junctions [123]) in keratinocyte cytoskeleton in a time-dependent manner, this means that luteolin-7-glucoside upregulates this protein. Also, cells treated with this flavone (50 ng/mL) promoted lipid raft generation and decreased fatty acid β-oxidation, methionine sulfoxide, and oxidized glutathione levels in HEKn cells. On the other hand, luteolin-treated cells increased the cortisol level and reduced levels of PGE_2_. Together these results confirm the antioxidant and anti-inflammatory effects of luteolin-7-glucoside in keratinocytes. The authors stimulate HEKn cultures with IL-22 because this cytokine is a key cytokine in psoriasis. It can induce regenerative and proliferative processes, inhibit differentiation, and have pro-inflammatory properties in keratinocytes. Treatment with this flavone (0.4 or 4 mM) reverts the expression of KRT10 and KRT1 and counteracts the effect of IL-6 on KRT10 differentiation and expression. In addition, IL-22 induces STAT3 phosphorylation; this is related to the induction of acanthosis (keratinocyte proliferation) and dermal inflammation in psoriasis. Nevertheless, treatment of HEKn cells with luteolin-7-glucoside did not have an effect on STAT3 phosphorylation, although it appeared to block IL-22 translocation to the nucleus of keratinocytes. In addition, topical treatment in imiquimod-induced psoriasis-like lesions on the mice model with luteolin-7-glucoside reduced the presence of STAT3 in the nucleus of skin cells. This means that the flavonoid can modulate the expression of STAT3, blocking the nuclear translocation of the transcription factors induced by IL-22 [122]. In this same research, topical treatment with luteolin-7-glucoside reduced skin-like psoriasis lesions because the histological evaluation of C57BL/6 mice decreased epidermal and scale thickness. Moreover, keratinocyte proliferation markers Ki67 and p63 decreased, and, additionally, the differentiation markers KRT10, transglutaminase-1 (TGase-1), and Loricrin were increased with topical treatment with the flavone (0.4 or 4 mM), which suggested that this flavonoid reverts the psoriasiform phenotype among inflammatory and proliferative responses in a dose-dependent manner on in vivo mice model [122].

**Table 6 antioxidants-10-02014-t006:** Beneficial properties of flavones in different psoriasis models.

Flavones	Model/Psoriasis Inducer	Activities	Ref.
Luteolin	NHEKs and HaCaT cells	Inhibits mRNA expression of IL-8 and IL-6; reduces mRNA expression of NFKB1 and RELA and HaCaT cell proliferation	[120]
	HaCaT cells	Inhibits transcriptional expression of HSP90β and HSP90α; decreases exosomes amount	[121]
	BALB/c mice/ imiquimod	Reduces psoriatic area, PASI, histological damage, and HSP90β and HSP90α expression; decreases the proportion of Th17/Treg cells and Th1/Th2	
	Raw264.7 macrophages	Decreases both mRNA and protein levels of TNF-α, IL-1β, IL-6, IL-23, and IL-17A; inhibits NF-κB p65 and COX-2 expression; suppresses iNOS and NO	[39]
	BALB/c mice/ imiquimod	Reduces PASI, infiltration of macrophage, T cells, and neutrophil, and IL-1β, IL-17A, IL-23, and IL-6 expression	
Luteolin-7-glucoside	HEKn	Modifies cell cycle, energy, fatty acid, and redox metabolism; increases KRT10 expression and cortisol levels; decreases PGE_2_ level	[122]
	C57BL/6 mice/imiquimod	Reduces skin psoriasis-like lesions, Ki67 and p63; increases KRT10, TGase-1, and Loricrin; modulates expression of STAT3	
Baicalein	HaCaT cells	Inhibits cell growth; causes growth arrest in G_0_/G_1_phase; induces morphological differentiation Ca^2+^ dependent; increases KRT10 and KRT1 expression and ERK phosphorylation; activates TRPV4	[123]
Baicalin	BALB/c and ICR mice/DNFB	Reduces psoriatic symptoms, edema, and inflammatory cell infiltration; increases orthokeratosis	[124]
	BALB/c mice/imiquimod	Reduces psoriatic symptoms, PASI, histological damage, expression of TNF-α, IL-23, IL-22, and IL-17A, and γδT cells infiltration	[116]
Chrysin	NHEKs	Attenuates phosphorylation of JNK, ERK, and p38 kinase; decreases p-STAT3 and p-JAK2; suppresses CCL20 and AMPs expression	[35]
	BALB/c mice/imiquimod	Reduces psoriatic symptoms, PASI, histological damage, TEWL, erythema, blood flow, and thickness; increases the content of surface skin hydration	
Tangeretin	HaCaT cells	Inhibits the nuclear translocation of NF-κB p65 and HIF-1α	[125]
	BALB/c mice/PMA-induced ear inflammation	Reduces psoriatic symptoms, histological damage, PMA-induced hyperplasia; down-regulates the production of IL-1β, IL-6, IL-4, TNF-α, IFN-γ, PGE_2_, COX-2, MIP-2, MCP-1, KC, TRX, nNOS, iNOS, eNOS, MMP-9, MMP-2, TLR4, VEGF, p-Akt, p-p38, p-JNK, p-ERK1/2, NF-κB p65, NF-κB p50, IκBα, IKK-γ, and HIF-1α; suppresses MDA, and NO expression; increases SOD-2, HO-1, and CAT expression	

Growth of HaCaT cells treated with baicalein (10 or 30 μM) was altered, this arrested the cell cycle in the G_0_/G_1_ phase, whereas that morphological differentiation induced by the flavonoid was Ca^2+^ dependent. On the other hand, baicalein did not increase ROS or alter the integrity of the mitochondrial outer membrane; therefore, baicalein did not activate the apoptotic mitochondrial-related pathway in keratinocytes. Morphological changes in baicalein-induced in keratinocytes were associated with increases in KRT10 and KRT1 expression. This process is regulated by the ERK phosphorylation, which is dependent on the activation of the transient receptor potential V4 (TRPV4), which is activated by baicalein. In summary, baicalein decreased the proliferation of keratinocytes and accelerated the differentiation of these cells via activation of TRPV4, which could be an approach to the treatment of psoriasis [123].

Baicalin, also known as baicalein-7-glucuronide [14], has been evaluated in 2,4-dinitro-fluorobenzene (DNFB)-induced contact hypersensitivity in mice and mouse tail models. In both psoriatic-induced murine models, baicalin (1%, 3%, or 5% baicalin cream) reduced alterations in the skin produced by DNFB to reduce ear thickness and ear weight, thymus index, and spleen index. It was also reported that baicalin produced differentiation in the epidermis because of baicalin-induced keratinocyte differentiation, which is related to the enhancement of orthokeratosis and the drug activity of the flavonoid tested in a dose-dependent manner; also, the anti-inflammatory effects of baicalin were observed to reduce tissue edema and inflammatory cell infiltration in the skin treated with this flavonoid [124]. In vivo anti-psoriatic activity of baicalin (1.4 mg) was tested in BALB/c mice where psoriatic lesions were induced with imiquimod cream. An increase in psoriatic lesion after baicalin topical treatment was reported, including reduction of redness, scaling, and thickness, as well as the attenuation of the inflammatory response used to determine PASI in the treated mice and reversal of the loss of the granular layer induced by the imiquimod cream in mice. Cytokine analysis of inguinal lymph node cells revealed that baicalin treatment reduces the expression of TNF-α, IL-23, IL-22, and IL-17A; and immunohistochemical analysis showed the reduction of dermal γδ T cells (associated with the production of TNF-α, IL-22, and IL-17) after treatment with the flavonoid [116].

In 2020 Li et al. [35] found that the in vitro evaluation of chrysin did not display cytotoxic effects on NHEKs at 30 μM; however, this flavonoid at 50 μM affects NHEKs viability and then produces cytotoxicity. Stimulation of NHEKs with IL-22, IL-17A, or TNF-α increased phosphorylation of JNK, ERK, and p38 kinase, all components of the MAPK; all of which were attenuated by chrysin (3, 10, or 30 μM), moreover, p-STAT3 and p-JAK2 IL-22-induced expression and IκBα (the α isoform of IκB in the NF-κB signaling pathway) IL-17A- or TNF-α-mediated induction also was inhibited. Additionally, IL-22, IL-17A, or TNF-α upregulates the expression of chemokines such CCL20 and AMPs, including IL-37, human β defensin 2 (hBD2), S100 calcium-binding protein A9 (S100A9), S100A8, and S100A7 in NHEKs, which were suppressed by chrysin treatment. A different approach, such as the in vivo study of chrysin (30 mM), showed that it reduced the damage and PASI, and also reduced TEWL, erythema, blood flow, and ear thickness, while, on the other hand, increasing surface skin hydration [35].

Chang et al., in 2020 [125], used HaCaT cells to evaluate the anti-psoriatic potential and mechanisms involved for flavonoids, such as tangeretin, which did not exhibit cytotoxic effects or induce morphological or cell density alterations at a 50 μM concentration; additionally, it inhibited the translocation of NF-κB p65 subunit and hypoxia-inducible factor 1α (HIF-1α). In this same investigation, in phorbol 12-myristate 13 acetate (PMA)-induced ear inflammation, tangeretin reduced the ear thickness, edema, and hyperplasia induced by PMA. Histological analysis showed that treatment with tangeretin (10 or 39 mg/kg) reduced hyperkeratosis, the influx of inflammatory cells, granulation tissue accumulation, re-epithelialization, and epidermal hyperplasia and, moreover, suppressed serum IgE as well as downregulating the production of IL-1β, IL-6, IL-4, TNF-α, IFN-γ, PGE_2_, COX-2, macrophage inflammatory protein-2 (MIP-2), monocyte chemoattractant protein-1 (MCP-1), and keratinocyte chemoattractant (KC), which are markers of inflammation. It was also shown to downregulate the protein expression of thioredoxin (TRX), neuronal nitric oxide synthase (nNOS), iNOS, endothelial nitric oxide synthase (eNOS), and suppress the expression of MDA and NO, which are oxidative stress markers. Furthermore, it increased the protein expression of the antioxidant enzyme SOD-2, HO-1, and CAT. In this in vivo model, tangeretin induced systematic downregulation of MMP-9, MMP-2, TLR4, vascular endothelial growth factor (VEGF), p-Akt, p-p38, p-JNK, and p-ERK1/2 markers of cell proliferation. Finally, NF-κB p65, NF-κB p50, IκBα, IκB kinase-γ (IKK-γ), and HIF-1α hypoxia markers, and NF-κB pathway markers were dose-dependent downregulated by tangeretin [125]. Of general form all flavones reported with effects related to the anti-psoriatic activity were tested on keratinocytes cell lines in which most of the flavones [35,120,121,122,123,125] inhibit the cell growth of these cells; this property that displays the compounds tested it is through distinct mechanism that involved the inhibition of different factors of transcription or arrest of the cellular cell cycle. Moreover, in vivo evaluation of flavones decreases the psoriatic area in both histological and PASI measurements (Table 6). In this line, it becomes clear that the reduction or inhibition of key cytokines in the pathogenic mechanism of psoriasis (NF-κB, TNF-α, IL-1β, IL-23, and IL-17) plays an important role in the activity of the flavones. Although each compound presents different forms and has different activity on chemokines, cytokines, transcription factors, and molecules related to the development of psoriasis.

### 3.2. Flavonols

Quercetin is one example of a flavonoid with anti-psoriatic activity; the drug effect was determined topically in a murine model (25 or 50 mg/kg). In other words, when the increase in the percentage of orthokeratosis in the zone treated with quercetin was evaluated, this secondary metabolite was found to have antiproliferative activity in HaCaT cells, usually used in psoriasis as a model of epidermal hyperproliferation. In this study, quercetin changed the epidermal thickness of mouse tail skin treated topically with this flavonoid. It also showed a decrease or absence of the granular layer of the epidermis in psoriatic lesions treated with quercetin. Is it important to mention that the basis of the mouse tail test (used for the evaluation of the anti-psoriatic activity of quercetin) is the induction of orthokeratosis. Furthermore, quercetin has anti-inflammatory activity similar to the effects of COX antagonists, such as indomethacin, and COX-2 antagonists, such as celecoxib. This suggests that this flavonoid could modulate the inflammatory response in psoriasis [115] (Table 7). In another study related to the anti-psoriatic activity of quercetin, this flavonoid reduced the psoriatic damage induced by imiquimod in a murine model with male BALB/c mice, in which the skin of psoriatic mice treated with quercetin (30, 60, or 120 mg/kg) ameliorated skin erythema, scaling, and thickness; these signs were measured to determine the PASI, which is used to monitor and grade the severity of the psoriatic-like lesions [126]. Also, the temperature of the psoriasis-like lesions was reduced to nearly that of the healthy mice, and the histological analysis of psoriatic skin of mice treated with quercetin showed a smoother epidermis and less parakeratosis and epidermal thickening. This is an important finding because imiquimod induced inflammation in the mouse model, which shows similar symptoms of psoriasis in humans. Topical application of imiquimod on the skin of mice recruited large amounts of immune cells and active dendritic cells and induced the secretion of inflammatory cytokines and increased hyperplasia of the epidermis [122,126]. Treatment with quercetin levels of SOD, CAT, and GSH antioxidant markers was increased, and, therefore, lipoperoxidation in the psoriatic tissue was decreased. There was also a decrease in the production of IL-17, IL-6 and TNF-α pro-inflammatory cytokines and a reduction in the expression of the transcription factor RelB, IKKα and NF-κB inducing kinase (NIK). Additionally, there was an increase in the expression of TNFR associated factor 3 (TRAF3). Taking these results together, on the one hand, quercetin could regulate the oxidative/anti-oxidative status to a more favorable physiological equilibrium in mice with psoriasis-like lesions, and, on the other hand, the inflammatory status of psoriatic mice was improved favorably after the treatment with quercetin, because of the capacity of quercetin to inhibit the activation of NF-κB through canonical and non-canonical NF-κB signaling [126].

A research group reported that fisetin inhibits the cell proliferation of NHEKs, HaCaT, and A431 (epidermoid carcinoma cell line), of which NHEKs were less sensitive to the flavonoid tested; nevertheless, 5–30 μM did not show toxicity in the cell lines used. It should be noted that fisetin at 20 μM did not induce apoptosis in any of the cases; as evidence of this, caspases 9, 8, and 3 and/or changes in Bcl2, Bax, Bak, and PARP protein levels did not change. On NHEKs, fisetin dose-dependently induced TGase, a marker of terminal keratinocyte differentiation, and increased the expression of TGase-1, filaggrin, caspase-14, KRT10, and KRT10 differentiation markers. This flavonoid also increased the nuclear expression of members of AP-1 (a transcription factor that plays an important role in the regulation of keratinocyte differentiation, terminal differentiation, cytokine production, and inflammation) factor subunits Fos (Fos B, c-Fos, and Fra-1/2) and Jun (JunD, JunB, and c-Jun) [127]. NHEKs treated with fisetin suppressed the IL-22-induced proliferation through the P13K/Akt/mammalian Target of Rapamycin (mTOR) signaling pathway, as well as TNF-α-induced activation of MAPK and P13K/Akt/mTOR signaling pathways. Secretion of TNF-α, IL-1β, IL-1α, IL-8, IL-6, and the profibrotic mediator TGF-α induction in NHEK cells by 12-O-tetradecanolylphorbol 13-acetate (TPA) were reduced by fisetin, moreover, in peripheral blood mononuclear cells (PBMCs) inhibited IL-17A and IFN-γ mRNA accumulation. In this research, a 3D-full-thickness human skin model of psoriasis (FTRHSP) topically treated with fisetin (10–20 μM) showed suppression of proliferation and induction in the expression of differentiation markers desmoglein-1, TGase-1, filaggrin, involucrin, and KRT10 in the spinous via granular epidermal cell layers, and suppressed the expression of IL-17A, fosfo-p70 ribosomal protein S6 kinase (p-p70S6K), and psoriasin markers related to the activation of mTOR pathway [127].

Similarly, kaempferol at a 6 or 12 μM concentration suppressed T cell proliferation, although this concentration was not cytotoxic in T cells. Interestingly, it inhibited the phosphorylation of p70S6K downstream of the mTOR signaling [117]. Moreover, these same authors mention that oral treatment with kaempferol (50 or 100 mg/kg) reduced the severity of psoriatic-like lesions induced by imiquimod cream and that PASI reduction was closely related to CD4^+^CD25^+^Treg cells. Skin histology of treated male BALB/c mice showed smoother epidermis and reduced parakeratosis; immunohistochemistry analysis revealed a reduction in CD3^+^ T cells infiltration in the affected skin after kaempferol treatments; furthermore, this flavonoid increased the frequency of CD4^+^FoxP3^+^Treg cells in lymph nodes and spleens of treated mice. On the other hand, kaempferol reduced the percentage of IL-17A^+^CD4^+^ or RORγt^+^CD4^+^ T cells in lymph nodes and spleens, and also reduced mRNA expression of TNF-α, IL-6, and IL-17A while upregulating gene expression of IL-10 and FoxP3 and even inhibits the expression of phosphorylated NF-κB p65 [117]. To difference with the flavones, the studies related to the flavonols present different ways to evaluate their anti-psoriatic effects (Table 7). Both quercetin and kaempferol reduce the psoriatic symptoms in the murine imiquimod-induce skin damage like psoriasis model, and although both flavonoids share the decrease of TNF-α, IL-6, and IL-17 (cytokines related with psoriasis) the mechanisms involved in the activity of both compounds it is different because quercetin is related with their antioxidant properties and their capacity to regulates the expression of key molecules in psoriasis, and, on the other hand, the anti-psoriatic activity of kaempferol is closely related with the modulation of inflammation in the psoriatic tissue. Finally, only fisetin did not test on in vivo experimental model; nevertheless, displays different properties related to the inhibition of the cellular growth, signaling, and inflammatory profile of different cell lines used to test the mechanism of action that lead to the development of psoriasis, and the properties of fisetin could be used to determine their possible anti-psoriatic activity on in vivo experimental models.

### 3.3. Flavanones, Flavan-3-ols, and Isoflavones

Previously published papers evaluated the anti-psoriatic activity of naringenin and encapsulated naringenin with sericin. In this study, both naringenin alone and encapsulated naringenin (3.1 to 200 μg/mL) inhibited TNF-α production in human PBMCs (hPBMCs). Encapsulated naringenin seemed to be more active than naringenin alone, therefore, in this context, the inhibition of TNF-α blocked their activity, and this could reduce the interaction among the immune system and keratinocytes in psoriasis. Finally, this work showed that encapsulated naringenin is more effective than naringenin alone [128] (Table 8). In line with this, in 2020, Alalaiwe et al. [8] tested the in vivo anti-psoriatic activity of naringenin on imiquimod-induced psoriasis-like skin lesions in female BALB/c mice, and an increase in the cutaneous absorption capacity and availability at the site of action on the part of this flavanone was observed. In addition, the naringenin reduced the symptoms of psoriasis and the PASI test. Histological analysis revealed that this flavanone showed improvement in psoriasiform plaque and epidermal thickness and decreased the TEWL. Additionally, the naringenin inhibited IL-6, neutrophil migration, and keratinocyte proliferation, suggesting that the naringenin had anti-psoriatic activity, especially in the recovery of the cutaneous barrier.

Similarly, hesperidin (5, 10, or 20 μg/mL) inhibited LPS-induced keratinocytes cell proliferation, this work also reported the downregulation of protein expression of p-ERK1/2 in a dose-dependent manner. In addition, this flavanone regulated the metabolism of HaCaT cells [129]. In this same investigation, hesperidin reduced the damage, PASI, and histological changes induced by imiquimod in male BALB/c mice, although experimental animals were treated orally with different doses of flavanone tested (125, 250, or 500 mg/kg). In this work, hesperidin reduced the number of epidermal cells on the basal layer, the number of mitotic cells, total nuclei, and the ratio of mitotic positive cells, as well as involucrin expression level with decreased expression areas. IL-22, IL-23, IL-17, TNF-α, and IL-1β mRNA expression levels were decreased, although IL-6 mRNA was not suppressed by hesperidin. Moreover, p-ERK1/2 level and ratio were also reduced; leptin and adiponectin were increased while the resistin level was decreased by hesperidin treatment [129].

**Table 8 antioxidants-10-02014-t008:** Main anti-psoriatic effects of flavanones, flavan-3-ols, and isoflavones in different study models.

Flavonoids	Model/Psoriasis Inducer	Activities	Ref.
Naringenin	hPBMCs	Suppress LPS-induced serum TNF-α levels	[128]
	BALB/c mice/imiquimod	Reduces psoriatic symptoms, PASI, histological damage, and TEWL; inhibits neutrophil migration and IL-6	[8]
Hesperidin	HaCaT cells	Inhibits LPS-induced cell proliferation; down-regulates expression of p-ERK1/2	[129]
	BALB/c mice/imiquimod	Reduces psoriatic symptoms, PASI, histological damage, expression of involucrin, IL-22, IL-23, IL-17, TNF-α, and IL-1β; decreases p-ERK1/2 level	
EGCG and EGCG nanoparticles	NHEKs	Induces differentiation; increases involucrin, TGase-1, KRT10, and caspase-14; inhibits CXCL2, TGF-β, TNF-α, IL-8, IL-6, and IL-1β	[130]
	BALB/c mice/imiquimod	Reduces psoriatic symptoms and loricrin expression; decreases Ki67 expression, infiltration CD4^+^ T lymphocytes, and tissue vascularization; restores JunB and KRT10 expression; inhibits expression of IL-1β and TNF-α	
EGCG		Reduces psoriatic symptoms, PASI, histological damage, lipoperoxidation; decreases CD4^+^ T cells infiltration, IL- 23, IL-22, IL-17F, and IL-17A levels; increases CAT and SOD bioactivities	[131]
Genistein	HaCaT cells	Decreases MCP-1, VEGFA, TNF-α, IL-23, IL-8, and IL-1β; inhibits IκBα phosphorylation; decreases NF-κB level	[132]
	BALB/c mice/imiquimod	Reduces histological damage, CD45 inflammatory cell infiltration, TNF-α, IL-6, IL-1β, CCL2, IL-23, and IL-17; suppresses STAT3 phosphorylation	

Other work related to the in vitro and in vivo determination of EGCG psoriasis healing activity reported that the encapsulation of this flavonoid in polymeric nanoparticles enhanced their properties. In vitro anti-psoriatic evaluation (0, 5, 10, 20, 40, or 80 μM expressed as free EGCG equivalents) showed that EGCG nanoparticles inhibit cell growth/viability of NHEKs in a dose- and time-dependent form at a 2–4-fold lower concentration than free EGCG. In line with this, EGCG nanoparticles inhibit IL-22-induced proliferation on NHEKs at 4-fold less concentration than free EGCG. The encapsulation of EGCG enhanced the pro-differentiative properties of NHEKs of this flavonoid because EGCG nanoparticles at 5 μM induce the expression of involucrin, TGase-1, KRT10, and caspase-14 which are markers of keratinocyte differentiation, and are deregulated in inflamed psoriatic lesions, with respect to free EGCG that displays their activity at 20 μM. This same behavior is shown in the in vitro TPA-induced model of keratinocyte inflammation where EGCG nanoparticles (5 μM) and free EGCG (20 μM) were tested. In this model, both forms of EGCG tested inhibit CXCL2, TGF-β, TNF-α, IL-8, IL-6, and IL-1β [130]. Moreover, in this same research, the topical evaluation of free EGCG (1 mg/cm^2^ shaved skin/right ear area) and EGCG nanoparticles (48 μg/cm^2^ shaved skin/right ear area) in the imiquimod-induced psoriatic lesion in BALB/c mice showed that both forms of EGCG suppressed psoriasis-like symptoms at both the macroscopical and histological level, although EGCG nanoparticles were even more successful in blocking erythema. Also, EGCG nanoparticles display anti-hyperproliferation of keratinocyte activity by decreasing expression of Ki67^+^-epidermal keratinocytes/dermal. Both forms of EGCG decreased infiltration of leucocytes in skin psoriatic-like lesions, including CD4^+^T lymphocytes. Different markers that are involucrate in the differentiation of keratinocytes as KRT10, filaggrin, caspase-14, and loricrin, as well as JunB (that release inflammatory mediators that contribute to the psoriatic pathology) also were tested; in all cases, both forms of EGCG had activity on these markers; moreover, EGCG nanoparticles had better effects than free EGCG. Finally, both forms of EGCG strongly decreased tissue vascularization, and pro-inflammatory cytokines IL-1β and TNF-α [130].

In other research, EGCG (150 or 300 mg/kg) reduced erythema, scales, and infiltration from the second day of treatment in lesioned skin of BALB/c mice with imiquimod cream to produce psoriasis-like skin damage. This was reflected in the reduction in PASI with respect to mice untreated with EGCG. This tendency was replicated in the histological analyses of mice skin, which did not show significant differences with respect to normal mouse skin. EGCG reduced infiltration of T cells, and it should be noted that this flavonoid reduced the splenomegaly induced by imiquimod topical treatment. Also, it was reported that the increase of CD4^+^ T cells further reduced the percentage of CD11c^+^ dendritic cells, and had no effects on CD8^+^ T cells in spleens. With respect to the inflammatory cytokines involucrate in the development of psoriasis, EGCG treatment reduced the level of IL-23, IL-22, IL-17F, and IL-17A. The activity of antioxidant enzymes CAT and SOD was enhanced, which was reflected in the reduction in lipoperoxidation of the mouse skin treated with the flavonoid [131]. It was reported that genistein (50 or 100 μM) inhibited the proliferation of TNF-α-treated HaCaT cells, in these, decrease the content of MCP-1, VEGFA, TNF-α, IL-23, IL-8, and IL-1β, and also inhibited phosphorylation of IκBα and decreased the level of NF-κB [132]. In this same study, the genistein cream (0.5% or 2%) enhanced the skin damage. Imiquimod-induced, histological damages were diminished in a dose-dependent manner, CD45 inflammatory cell infiltration was also reduced, as well as Th1 cytokines such as TNF-α, IL-6, IL-1β, and Th-17 cytokines such as CCL2, IL-23, and IL-17. Also, this isoflavone suppressed phosphorylation of STAT3 in a concentration-dependent manner, presenting an anti-TNF-α-induced STAT3 phosphorylation effect [132]. Although the compounds tested belong to different subgroups of flavonoids, all of them have anti-psoriatic properties both in vitro and in vivo. It is interesting that, too, despite the differences among them all compounds have the capacity to reduce the psoriatic symptoms to histological level and have in common that the possible mechanisms that present are related to the regulation of inflammatory process in the damaged tissue by psoriasis. It is important to denote that only the anti-psoriatic activity of EGCG is related with their antioxidant activities [131], unlike the flavanones as naringenin, and hesperidin, or the isoflavone genistein that also have strong antioxidant properties [49,87]. Therefore, the research of the antioxidant effects of these subgroups of flavonoids should be explored in future works to have a better understanding of the anti-psoriatic activities of these flavonoids both in vivo and in vitro experimental models.

As mentioned in this section, flavonoids have various properties that combat psoriasis, among which are their ability to increase antioxidant enzymes; to reduce psoriasis symptoms, the PASI index, lipoperoxidation, various pro-inflammatory cytokines, mediators of inflammation, and oxidative stress molecules; and also to regulate the cell cycle, as well as some signaling pathways, even at the genic level. Lately, studies on the efficacy of the release and topical permeability of nanoparticles and hydrogels containing different flavonoids, such as naringenin, quercetin, or luteolin, have been implemented to observe and determine the effectiveness of these new formulations in the treatment of psoriasis [133,134].

It is important to comment that, so far, no clinical studies have been started based on topical treatment with flavonoids in patients who have psoriasis. However, some working groups have mentioned the activity of topical treatment with a cream formulated with propolis and *Aloe vera* in patients with psoriasis. The authors observed that it reduces psoriatic lesions, confirming the above through histological analysis. In addition, they attribute the effectiveness of the cream to flavonoids that could be present in the chemical composition of propolis and *A. vera* [9,135]. Nevertheless, these authors do not report the components that make up propolis or *A. vera*, and, if it could be reported, it would help to know the bioactive compounds of these products and, thus, complement these researches. Interestingly, so far, there are no basic research works (in vitro or in vivo) where propolis is used as an anti-psoriatic agent. This is worrying since it is necessary to strengthen the knowledge bases to carry out more clinical studies with therapeutic approaches better directed.

Considering the above, the next step that needs to be performed in the in vitro and in vivo investigations of the activities of flavonoids on psoriasis described in this section is to implement clinical studies that promote the knowledge of the potential that these compounds have for the treatment of psoriasis.

If we analyzed the flavonoids that have been tested both photoaging and psoriasis found that some of them share properties that could be used in both pathologies; in the different subgroups of flavonoids, we found compounds that have been tested in parallel studies according to know their anti-photoaging and anti-psoriatic properties. Flavones and flavonols are the subgroups that present the majority of compounds tested in both illnesses (Table 1, Table 2, Table 6 and Table 7); moreover, both types of flavonoids also share some compounds tested in both pathologies such as the flavones luteolin [39,51,53,54,55,120,121,122], and chrysin [35,58] and the flavonols kaempferol [70,117], quercetin [70,74,76,115,122,126], and fisetin [81,127]. Although in the different studies in which these flavonoids have been evaluated the aims are different, some in vitro models used are similar between them; and it is in this point that we found the anti-inflammatory properties on keratinocytes cell lines that display to reduce the levels and regulate the activation, and/or expression of different molecules involved in both photoaging and psoriasis [51,53,54,55,58,120,121,127]; therefore, this mechanism should be taken into account, and to take advantage of the antioxidants properties of the flavonoids tested on photoaging models [38,50,53,55,58,81] to be applied in psoriasis models. Finally, the flavanones hesperidin [83,84,129], and naringenin [8,85,86,128]; the flavan-3-ol EGCG [88,89,90,130,131]; and the isoflavone genistein [97,132], also are tested in both illnesses; although it should be mentioned that the number of flavonoids studied of these subgroups is lower in comparison with the flavones and flavonols; therefore, ranking to evaluate their anti-photoaging, and anti-psoriatic properties becomes difficult to establish because there are so many signaling pathways that are not taken into account on in vitro and in vivo experimental models of both pathologies.

As already described, the antioxidant activity presented by each flavonoid is correlated with the number of hydroxyl substitutions it contains in its structure, and ether/acetal bonding (O-methylation or O-glycosylation) of the hydroxyl substitution reduces this activity [136,137,138]. And, probably, these aspects also coincide in the different properties they present, such as their anti-inflammatory, anti-photoaging, and anti-psoriatic activity, among others. It should be noted that the differences in the chemical structure of each flavonoid probably confer on it the different biological properties, presenting differences in the concentrations used and depending on the experimental model used. However, it is mainly the antioxidant activity that generates a positive effect in the regulation of various alterations both in photoaging as well as in psoriasis.

## 4. Conclusions

In this review, we focus on describing the role of the flavonoids present in propolis in the treatment of photoaging and psoriasis, conditions that have a great impact on the quality of life of the sufferer, can even cause more serious diseases, and lack adequate and accessible therapy.

With respect to photoaging, there is evidence that shows that flavonoids participate in the regulation and recovery of damage caused by UV radiation, such as DNA damage, inflammation, immunosuppression, cell senescence due to telomere shortening, and cell death. The reestablishment of the homeostasis of the flavonoids antioxidant system is linked to the regulation of the NF-κB, MAPK, STAT, JAK, Nrf2, and the transcription of stress response genes, such as GST, HO-1, and NQO-1. However, there are still important parameters to be described and studied, mainly in clinical studies with humans, such as their effects on the regulation of inflammasome, autophagy, and pyroptosis, as well as some immunological parameters, such as Langerhans cells, keratinocytes, Treg cells, and polarizes the Th1/Th2 response.

On the other hand, diverse flavonoids present in the chemical composition of propolis have anti-psoriatic activity; this activity is related to the reduction of inflammation through the decrease of different cytokines and mediators that include TNF-α, NF-κB, IL-1β, IL-17, Il-6, IL-22, IL-23, PGE_2_, and COX enzymes; the inhibition of transcriptional expression of HSP90 subunits; as well as the induction of their antioxidant activity to increase the activity of enzymes SOD, and CAT, and by increasing of GSH levels, which all together increases the orthokeratosis in damaged skin by psoriasis. Moreover, the antiproliferative effects of some flavonoids, such as quercetin, baicalein, fisetin, kaempferol, and genistein, could be important in the reduction of psoriatic skin damage. Finally, flavonoids can reduce psoriatic symptoms, both macroscopical to histological level, through different mechanisms are involved in the psoriatic process.

As we already mentioned, honeybee products and their derivatives, such as flavonoids, have been shown to have different health benefits, even though their use is increasingly regulated in countries such as the US and Canada. In addition, working and using pure bioactive compounds will make it easier to determine the specific dosage for each type of flavonoid and thus achieve a safe consumption level for people. Finally, some flavonoids have been shown to be effective in relieving other skin diseases, such as melasma and acne, and promoting healing processes, which is why bee products are a promising source of bioactive compounds that complement the treatment of different skin pathologies and even open the door to the study of other types of compounds, such as terpenes, that are also found in honeybee products.

## Figures and Tables

**Figure 1 antioxidants-10-02014-f001:**
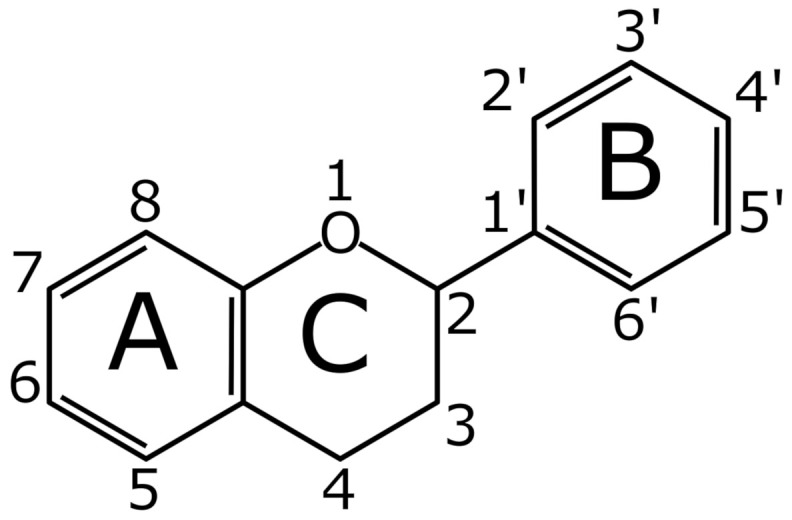
Basic flavonoid structure.

**Table 2 antioxidants-10-02014-t002:** Main anti-photoaging effects of flavonols in different study models.

Flavonols	Model/UV Radiation	Activities	Ref.
Quercetin, myricetin, and kaempferol	HDFs/UVA	All flavonoids inhibit MMP-1 mRNA levels; presents antioxidant activity	[50]
Galangin, kaempferol and quercetin	EpiDerm^TM^/UVA or UVB	All flavonoids reduce MMP-1 and TNF-α secretion; pretreatment reduces the cyclobutane thymine dimers in UVB-irradiation	[70]
Myricetin	PHKC/UVB	Decreases MDA level; suppresses H_2_O_2_ production and JNK activation	[71]
	Hairless mice/UVB	Decreases epidermal thickening; inhibits enzyme activity and MMP-9 protein expression; inhibits Raf kinase activity and, consequently, decreases MEK and ERK phosphorylation	[72]
Quercetin and Quercetin nanoparticles	HaCaT cells/UVB	Both decrease NF-κB protein; suppress the IkB-α phosphorylation; reduce the COX-2 protein expression level	[74]
	C57BL/6 mice/UVB	Decreases IkB-α phosphorylation, COX-2 expression, and PGE_2_ concentration	
Quercetin	PHKC/UVA and UVB	Reduces NF-κB DNA-binding; inhibits TNF-α, IL-1β, IL-6, and IL-8 expression	[75]
	Human abdominal skin tissue/UVA and UVB	Inhibits COX-2, MMP-1, and collagen degradation; suppresses NF-κB and AP-1 activation, and JAK2 and PKCδ kinase activity; decreases the Akt, JNK, ERK, and STAT3 phosphorylation	[37]
	Hairless mice/UVB	Decreases MPO activity, increases GSH, and suppresses proteinases secretion/activity	[76]
Quercitrin	JB6 cells/UVB	Suppresses apoptosis; reduces C-caspase-3 and C-PARP1 activation; decreases NF-κB activation, DNA damage, 8-OHdG production, γ-H2AX expression, and superoxide radical production; inhibits hydroxyl radical and hydrogen peroxide productions; increases SOD and CAT expressions	[77]
	Hairless mice/UVB	Decreases apoptosis, C-caspase-3 and C-PARP1 expression; DNA damage; 8-OHdG production; and γ-H2AX expression; increases XPA, SOD and CAT expression and GSH levels	
Fisetin	HDFs/UVB	Inhibits collagen degradation, MMP-9, MMP-3, and MMP-1 expression, COX-2, NO generation, the PGE_2_ and intracellular ROS, NF-κB translocation into the nucleus and the CREB phosphorylation level; decreases JNK and ERK expression, p38 phosphorylation, and IκB degradation	[81]

**Table 5 antioxidants-10-02014-t005:** Propolis from different countries and some of its identified flavonoids with anti-photoaging properties.

Propolis/Flavonoids	Model/UV Radiation	Identified Flavonoids	Activities	Ref.
Brazil (green propolis)	HDFs/ UVA	N.I.	Inhibits intracellular ROS generation, and ERK and p38 phosphorylation level	[109]
		N.I.	Positively modulates early HO-1 expression; induces the rapid translocation of Nrf2 to the nucleus	[105]
Iran	HDFs/ UVB	N.I.	Raises NGF and FOXO3A gene expression; reduces b-galactosidase activity; presents outstanding antioxidant activity	[110]
Korea/apigenin, and quercetin		Catechin, naringenin, apigenin, and quercetin	Propolis inhibits MMP-1 production, mRNA levels, and collagen degradation; suppresses Akt, PDK1, and PI3K activity; apigenin and quercetin inhibits PI3K activity	[106]
Korea	HaCaT cells/ UVA	N.I.	Suppresses apoptosis, C-caspase-3 expression; decreases the loss of mitochondrial membrane potential and ROS production	[111]
Greece	HaCaT cells/ UVB; EpiDerm^TM^/ UVB	N.I.	In HaCaT cells, presents antioxidant activity; reduces DNA damage and total protein carbonyl content. In EpiDerm^TM^, decreases MMP-9, MMP-7, MMP-3, and MMP-1 mRNA levels	[107]
Romania	Swiss mice/ UVB	Luteolin, kaempferol and apigenin	Reduces MDA and IL-6 levels, C-caspase-3 activation, sunburn cell formation, and CPDs generation; increases GPx activity	[108]

N.I. = No Identified.

**Table 7 antioxidants-10-02014-t007:** Anti-psoriatic activity of the flavonols present in propolis.

Flavonols	Model/Psoriasis Inducer	Activities	Ref.
Quercetin	Albino mice	Increases orthokeratosis; decreases granular layer of the epidermis; presents anti-inflammatory effects	[115]
	BALB/c mice/imiquimod	Reduces psoriatic symptoms, PASI, temperature of the psoriasis-like lesions, histological damage, IL-17, IL-6, and TNF-α; increases SOD, CAT, and GSH; downregulates the expression of RelB, IKKα, and NIK; upregulates TRAF3 expression	[126]
Fisetin	NHEKs, A431, HaCaT, PBMCs and FTRHSP	Inhibits cells growth of NHEKs, A431, HaCaT; on NHEKs induce TGase activity; increases nuclear expression of AP-1 factor subunits (factor Fos (Fos B, c-Fos, and Fra-1/2) y Jun (JunD, JunB, and c-Jun)); suppresses TNF-α-induced activation of MAPK and PI3K/Akt/mTOR pathway; reduces TNF-α, IL-1β, IL-1α, IL-8, IL-6, and TGF-α; in PBMCs inhibits IL-17A and IFN-γ mRNA accumulation; in FTRHSP suppresses expression of desmoglein-1, TGase-1, filaggrin, involucrin, and KRT10, IL-17A, p-p70S6K, and psoriasin markers	[127]
Kaempferol	T cells	Suppresses T cells proliferation; inhibits phosphorylation of p70S6K downstream of the mTOR signaling	[117]
	BALB/c mice/imiquimod	Reduces psoriatic symptoms, histological damage, PASI, CD3^+^T cell infiltration, IL-17A^+^CD4^+^ or RORγt^+^CD4^+^ T cells, mRNA expression of TNF-α, IL-6, and IL-17A; inhibits p-NF-κB p65 expression; promotes CD4^+^FoxP3^+^ Treg generation	

## Data Availability

Not applicable.

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
