# Peer review of "Flavonoids Present in Propolis in the Battle against Photoaging and Psoriasis"

_antioxidants, 2021, doi:10.3390/antiox10122014_

Round 1
Reviewer 1 Report
Review about effects of flavonides orginated from propolis on photoaging and psoriasis. it is interseting and actual topic.
Some sentences need rewriting, e.g. propolis is produt made by bees...- is not correct in whole (please, write correct definition of propolis).
In whole manuscript writing of abbreviations must be edited: firstly introduce term with full name and abbreviation in breckets....and every next time just abbreviation - e.g. L 100, 101, 102, 105, ....
in vivo, in vitro - write as italic - edit in whole manuscript
Introduction must be completed with more references regarding propolis properties , L 81, insert:
doi.org/10.3390/molecules26102930
doi.org/10.3390/antiox10060978
doi. 10.17221/103/2017-CJFS
In title of Table 4. - insert full stop
technical editing - alignment on both sides of text body
one work - replace with previously published articles..
histological exam??? - are You sure in this term?? - must be corrected or replaced with other phrase
beekeeping products - do you mean honeybe products?????
Author Response
Reviewer 1
Review about effects of flavonides orginated from propolis on photoaging and psoriasis. it is interseting and actual topic.
- Some sentences need rewriting, e.g. propolis is produt made by bees...- is not correct in whole (please, write correct definition of propolis).
Reply:
Regarding "Some sentences need rewriting, e.g. propolis is produt made by bees ...- is not correct in whole (please, write correct definition of propolis)."
In line 81 the following paragraph was added:
“natural resinous product that bees make using material obtained from multiple botanical sources; it is mixed with beeswax and enzymes secreted by bees through their salivary glands [14]. Propolis is a sticky, flexible and soft resin at warm temperatures, or brittle and hard at cold temperatures, it also presents various colors such as red, green or brown [15, 16]. Generally, the chemical composition of propolis is 50% resin, 30% wax, 10% essential oils, 5% pollen and 5% other substances [17]. This natural product has been reported to present approximately 300 different compounds [18]. The characteristic chemical groups characterized in propolis are flavonoids, steroids, phenolic acids or their esters, terpenes, stilbenes, aromatic aldehydes and alcohols as well as fatty acids [18, 19]. It is also known that both the chemical composition and the biomedical effect of propolis have a very high variability according to the collection region, the sources of vegetation in the area and the seasons [20, 21].”
We decided to use a brief definition because we do not intend to delve too deeply about propolis but rather to use it as a starting point since we intend to focus the review on the flavonoids that have been reported in propolis. And the other reason is to avoid duplication, since we have defined it in other of our publications on propolis, as mentioned below:
- Rivera-Yañez, N .; Rivera-Yañez, C.R .; Pozo-Molina, G .; Méndez-Catalá, C.F .; Méndez-Cruz, A.R .; Nieto-Yañez, O. Biomedical properties of propolis on diverse chronic diseases and its potential applications and health benefits. Nutrients 2021, 13, 78.
- Rivera-Yañez, N .; Rivera-Yañez, C.R .; Pozo-Molina, G .; Méndez-Catalá, C.F .; Reyes-Reali, J .; Mendoza-Ramos, M.I .; Méndez-Cruz, A.R .; Nieto-Yañez, O. Effects of propolis on infectious diseases of medical relevance. Biology 2021, 10, 428.
- Rivera-Yañez, N., Rodriguez-Canales, M., Nieto-Yañez, O., Jimenez-Estrada, M., Ibarra-Barajas, M., Canales-Martinez, MM, & Rodriguez-Monroy, MA (2018). Hypoglycaemic and antioxidant effects of propolis of Chihuahua in a model of experimental diabetes. Evidence-Based Complementary and Alternative Medicine, 2018.
And in the line 373 the following text was added:
“…that bees make using material obtained from multiple botanical sources…”
- In whole manuscript writing of abbreviations must be edited: firstly introduce term with full name and abbreviation in breckets....and every next time just abbreviation - e.g. L 100, 101, 102, 105, ....
Reply:
All text was reviewed and corrected on lines 107, 140, 157, 188, 206, 208, 209, 221, 230, 238, 242, 251, 263, 272, 386, 553, 568, 570, 576, 628, 631, 636, 772, 773 and Table 2.
- in vivo, in vitro - write as italic - edit in whole manuscript
Reply:
It was reviewed the instructions for authors of the journal "Antioxidants", as well as the latest articles it has published on its website and these words do not require the italic format.
- Introduction must be completed with more references regarding propolis properties , L 81, insert:
- doi.org/10.3390/molecules26102930
- doi.org/10.3390/antiox10060978
- doi. 10.17221/103/2017-CJFS
Reply:
References were added on line 82 of the text.
- In title of Table 4. - insert full stop
Reply:
Corrected in text.
- technical editing - alignment on both sides of text body
Reply:
Corrected formatting in all text.
- one work - replace with previously published articles..
Reply:
It was replaced by "Previously published papers" in the line 655 of the text.
- histological exam??? - are You sure in this term?? - must be corrected or replaced with other phrase
Reply:
The term "histological exam" was corrected. In line 666 was replaced by “histological analysis”.
- beekeeping products - do you mean honeybe products?????
Reply:
Yes, we mean honeybee products. In lines 789 and 798, beekeeping products, were replaced by “honeybee products”.

Reviewer 2 Report
- Line 229, The concentration of quercetin used as 20 mg/mL for keratinocytes or mice? Microgram or milligram, which is correct?
- Line 233, quercetin administration (5–40 M) on light UVA- and UVB-irradiated human abdominal skin tissue. This indicated that quercetin 1.5 mg/ml or 12 mg/ml. Which solvent in this study. Authors should be mentioned which solvent used in these studies for confirming their safety. In addition, authors must be made a table for discussing the concentration, used solvent and preparation methods of each study.
- For safety evaluation, the physical and chemical properties of propolis described too less, authors should be added the solubility, absorption. In addition, authors must also be mentioned in their safety in this review.
Author Response
Reviewer 2
- Line 229, The concentration of quercetin used as 20 mg/mL for keratinocytes or mice? Microgram or milligram, which is correct?
Reply:
The concentration is used for keratinocytes and the correct one is micrograms. 20 mg/mL was corrected for "20 µg/mL" in line 230 of the text.
- Line 233, quercetin administration (5–40 M) on light UVA- and UVB-irradiated human abdominal skin tissue. This indicated that quercetin 1.5 mg/ml or 12 mg/ml. Which solvent in this study. Authors should be mentioned which solvent used in these studies for confirming their safety. In addition, authors must be made a table for discussing the concentration, used solvent and preparation methods of each study.
Reply:
Regarding "Line 233, quercetin administration (5–40 M) on light UVA- and UVB-irradiated human abdominal skin tissue. This indicated that quercetin 1.5 mg / ml or 12 mg / ml."
The correct concentration is 5-40 µM. It was corrected in the text.
Regarding "Which solvent in this study."
In that study quercetin-sepharose 4B beads were prepared. Sepharose 4B powder was activated with 1 mM HCl and quercetin was conjugated to the activated Sepharose 4B beads in coupling solution by rotation overnight at 4º C.
Therefore, on line 234, quercetin administration was corrected for “administration of pearls containing quercetin”.
Regarding "Authors should be mentioned which solvent used in these studies for confirming their safety. In addition, authors must be made a table for discussing the concentration, used solvent and preparation methods of each study."
It is important to mention that in most of the investigations they do not specify some technical details, such as the solvent, so it is difficult to try to include this information in the review, since it would be incomplete. However, what is included in each of the studies mentioned in this review is the concentration of each flavonoid used. But it should be noted that there is a difference in terms of the concentration used of each flavonoid in the different in vitro and in vivo models.
For this reason, we add the following paragraph in the text in the line 761:
“As already described, the antioxidant activity presented by each flavonoid is correlated with the number of hydroxyl substitutions it contains in its structure, and ether/acetal bonding (O-methylation or O-glycosylation) of the hydroxyl substitution reduces this activity [136-138]. And probably these aspects also coincide in the different properties they present, such as their anti-inflammatory, anti-photoaging, anti-psoriatic activity, among others. It should be noted that the differences in the chemical structure of each flavonoid probably confer on it the different biological properties, presenting differences in the concentrations used and depending of the experimental model used. However, it is mainly the antioxidant activity that generates a positive effect in the regulation of various alterations both in photoaging as well as in psoriasis.”
Furthermore, we consider that the objective of our review is not to address the differences between the preparation method or the solvent used in each flavonoid, since all that information would be sufficient to publish a review on that topic, as can be found in these recent papers:
- Chaves, J. O., De Souza, M. C., Da Silva, L. C., Lachos-Perez, D., Torres-Mayanga, P. C., da Fonseca Machado, A. P., ... & Rostagno, M. A. (2020). Extraction of flavonoids from natural sources using modern techniques. Frontiers in Chemistry, 8.
- Šuran, J., Cepanec, I., Mašek, T., Radić, B., Radić, S., Tlak Gajger, I., & Vlainić, J. (2021). Propolis extract and its bioactive compounds — From traditional to modern extraction technologies. Molecules, 26 (10), 2930.
- For safety evaluation, the physical and chemical properties of propolis described too less, authors should be added the solubility, absorption. In addition, authors must also be mentioned in their safety in this review.
Reply:
Regarding "For safety evaluation, the physical and chemical properties of propolis described too less, authors should be added the solubility, absorption.”
In this review we do not intend to delve too deeply about propolis but rather to use it as a starting point since we intend to focus the review on the flavonoids that have been reported in propolis.
However, the following paragraph was added in the line 81 of the text:
“natural resinous product that bees make using material obtained from multiple botanical sources; it is mixed with beeswax and enzymes secreted by bees through their salivary glands [14]. Propolis is a sticky, flexible and soft resin at warm temperatures, or brittle and hard at cold temperatures, it also presents various colors such as red, green or brown [15, 16]. Generally, the chemical composition of propolis is 50% resin, 30% wax, 10% essential oils, 5% pollen and 5% other substances [17]. This natural product has been reported to present approximately 300 different compounds [18]. The characteristic chemical groups characterized in propolis are flavonoids, steroids, phenolic acids or their esters, terpenes, stilbenes, aromatic aldehydes and alcohols as well as fatty acids [18, 19]. It is also known that both the chemical composition and the biomedical effect of propolis have a very high variability according to the collection region, the sources of vegetation in the area and the seasons [20, 21].”
Also, we did not delve too deeply into the chemical composition of propolis, to avoid duplication, since we described this topic in another of our papers, as mentioned below:
- Rivera-Yañez, N .; Rivera-Yañez, C.R .; Pozo-Molina, G .; Méndez-Catalá, C.F .; Méndez-Cruz, A.R .; Nieto-Yañez, O. Biomedical properties of propolis on diverse chronic diseases and its potential applications and health benefits. Nutrients 2021, 13, 78.
So that the reader does not lose sight of the fact that the central theme of the review is flavonoids, we add on the line 350 the following paragraph, regarding absorption:
“In this regard, it is known that some types of flavonoids administered or applied directly do not show good biological activity, however, an effort has been made to improve the biological activity of flavonoids by investigating different forms of conjugation or com-bination with different substances such as sepharose beads, nanoparticles or gels based on microemulsions, sodium alginate or poly (vinyl) alcohol or even liposomes as amphipathic transporters of bioactive compounds, such as for example EGCG, quercetin, naringenin, among others [74, 88, 99-101]. An example of this is the research conducted by Esposito et al. in 2020 [99], who has been studied a novel series of hybrid hydrogels at different ratios based on sodium alginate and poly(vinyl) alcohol quercetin-loaded. The permeability of quercetin through the skin showed different penetration/permeation profiles according to the hydrogel’s blend, which will allow the selection of hydrogels ratio for a best local and prolonged skin effect, making the use of these hydrogels a promising solution for the delivery of flavonoids for the treatment of skin ageing and inflammation. Another example is the study carried out in 2020 by Parashar et al. [100], who mention that the topical delivery system for of sericin gel loaded with microemulsion that containing naringenin, displayed higher retention and deposition of naringenin in the skin. Moreover, this is a propitious topical delivery system for naringenin for pre-venting or inhibiting UV-induced skin aging, that displayed enhanced therapeutic potential when compared with plain naringenin. Other research mentions that liposomes are nanocarriers that are used to incorporate bioactive compounds or drugs to treat some skin diseases. Liposomes are membranes of different sizes (from micro to nanometers) mainly composed of cholesterol and phospholipids, forming structures similar to cell membranes. Also, they are composed of a lipid bilayer and an aqueous nucleus that gives them their amphiphilic property. Furthermore, to improve permeability in the skin, edge activators are added to the liposomes, which reduces the stiffness of the bilayer structure making it deformable. However, more research is needed regarding the efficacy of loaded liposomes to transport different bioactive compounds to study their release through the skin [101]. It should be noted that it is still a challenge to find and standardize the best method of administration or application of flavonoids in a safe and efficient way to achieve their best expected biological activity for clinical application.”
Regarding "In addition, authors must also be mentioned in their safety in this review.”
In the line 418, the following paragraph was added:
“Besides, it is important to highlight that there is very little information regarding the safety of propolis, since it is a subject of study little investigated, it was found that propolis is considered safe by Diniz et al. [114], based on the results obtained in the biochemical safety profile in your research. However, it is clear that more studies is needed on this subject, as well as on many of the natural products used as complementary treatments in various diseases. In addition, it is important to mention that it should start with the implementation of more investigations to establish and propose methods of analysis and extraction of the different compounds present propolis around the world, such as flavonoids, to be able to compare in a more efficient way the biomedical activities of the propolis and its bioactive components [23]. Furthermore, the little research on the safety and standardization of propolis are some of the limitations for its use in clinical practice.”

Reviewer 3 Report
This review addresses in detail the effect of flavonoids on keratinocytes, fibroblasts, photoaging and psoriasis. This is a great review, but some points should be addressed before publication:
- Because of its high grade of detail it is quite log. It gets even longer because there is a strict distinction between psoriasis and photoaging. In some cases this distinction is not justified, because between photoaging and psoriasis are some similarities (e.g. activation of MMPs). In my opinion it would be benefical to discuss the effect of flavoanoids on psorisasis and photoaging together. Pathways that are regulated are the same and therefore have an effect on both.
- This review also makes a distinction between flavones, flavonols, flavanones and isoflavones. Differences and joint chemical backbones should be explained, eventually a figure with chemical structures would help.
- In the introduction the impression is given that the review is about nutraceuticals. But most falvones have a very bad bioactivity and uptake. This is also shown by most experiments on mouse models described in this review, in which the flavanoids are topically applicated. Therefore I would suggest to make a chapter of its own, in which the best way of application is described. A good start is the section: line 340-370. It is compeletely unfitting at the current position (because it is not really linked to isoflavones), but already contains most statements
- Propolis should be explained in more detail. It should be stated that it is an undefined mixture of flavaonouds that differ between bee hives and therefore regions.
- It would be interesting to read what are the main plant sources of the flavanoids presented in this review
- Would it be possible to somehow rankt the flavanoids according their effect on photoaging and psoriasis?? I know that this is difficult…….but would be interesting
Minor points:
Line 63: Most skin diesaes have nothing to do with solar radiation
Line 92: By definition: Photaging is the result of chronic UVA and UVB exposure. In contrat to that skin aging is also cuased by different “agents”.
Line 95: Oxidative reactions lead to the activation of MMPs, MMPs are no oxidative reaction.
Conclusions: Some pathways (e.g. Keap-1) are for the first time meantioned in the conclusion section. A conclusion should be a conclusion ;-)
Author Response
Reviewer 3
This review addresses in detail the effect of flavonoids on keratinocytes, fibroblasts, photoaging and psoriasis. This is a great review, but some points should be addressed before publication:
- Because of its high grade of detail it is quite log. It gets even longer because there is a strict distinction between psoriasis and photoaging. In some cases this distinction is not justified, because between photoaging and psoriasis are some similarities (e.g. activation of MMPs). In my opinion it would be benefical to discuss the effect of flavoanoids on psorisasis and photoaging together. Pathways that are regulated are the same and therefore have an effect on both.
Reply:
In the line 761, the following paragraph was added:
“If we analyzed the flavonoids that have been tested both photoaging and psoriasis found that some of them share properties that could be used in both pathologies; in the different subgroups of flavonoids, we found compounds that have been tested in parallel studies according to know their anti-photoaging and anti-psoriatic properties. Flavones and flavonols are the subgroups that present the majority of compounds tested in both illnesses (Tables 1-2, and 6-7); moreover, both types of flavonoids also share some compounds tested in both pathologies such as the flavones luteolin [39, 51, 53-55, 120-122], and chrysin [35, 58] and the flavonols kaempferol [70, 117], quercetin [70, 74, 76, 115, 122, 126], and fisetin [81, 127]. Although in the different studies in which these flavonoids have been evaluated the aims are different, some in vitro models used are similar between them; and it is in this point that we found the anti-inflammatory properties on keratinocytes cell lines that display to reduce the levels and regulate the activation, and/or expression of different molecules involved in both photoaging and psoriasis [51, 53-55, 58, 120, 121, 127]; therefore, this mechanism should be taken in count, and to take advantage of the antioxidants properties of the flavonoids tested on photoaging models [38, 50, 53, 55, 58, 81] to be applied in psoriasis models. Finally, the flavanones hesperidin [83, 84, 129], and naringenin [8, 85, 86, 128]; the flavan-3-ol EGCG [88-90, 130, 131]; and the isoflavone genistein [97, 132], also are tested in both illnesses; although it should be mentioned that the number of flavonoids studied of these subgroups is lower in comparison with the flavones and flavonols; therefore, an ranking to evaluate their anti-photoaging, and anti-psoriatic properties it becomes difficult to establish because there are so many signaling pathways that are not taken into account on in vitro and in vivo experimental models of both pathologies.”
In the line 761, the following paragraph was added:
“As already described, the antioxidant activity presented by each flavonoid is correlated with the number of hydroxyl substitutions it contains in its structure, and ether/acetal bonding (O-methylation or O-glycosylation) of the hydroxyl substitution reduces this activity [136-138]. And probably these aspects also coincide in the different properties they present, such as their anti-inflammatory, anti-photoaging, anti-psoriatic activity, among others. It should be noted that the differences in the chemical structure of each flavonoid probably confer on it the different biological properties, presenting differences in the concentrations used and depending of the experimental model used. However, it is mainly the antioxidant activity that generates a positive effect in the regulation of various alterations both in photoaging as well as in psoriasis.”
- This review also makes a distinction between flavones, flavonols, flavanones and isoflavones. Differences and joint chemical backbones should be explained, eventually a figure with chemical structures would help.
Reply:
In the line 84, the following paragraph was added:
The flavonoids have a well-known chemical structure which consists of 15 carbon atoms that are arranged to form three aromatic rings named A, B, C. The B-ring is linked to the A-ring by a three carbon bridge that binds with one oxygen and two carbons of the A-ring thus forming the C-ring (Figure 1). Based on the different functional groups and level of oxidation in the C-ring, and different connections between B-ring and C-ring, flavonoids are classified into several groups such as flavones, flavonols, flavan-3-ols, isoflavones, flavanones, anthocyanidins, chalcones, and aurones [34].”
In the line 87 added "Figure 1. Basic flavonoid structure."
In the line 111, the following text was added:
“Flavones are a subclass of flavonoids; these compounds present in the C ring a ketone in position 4, and a double-bound between positions 2 and 3 [49].”
In the line 197, the following text was added:
“In simple terms, flavonols are flavonoids that have a ketone group; in comparison with the flavones, this subgroup presents in the C ring a hydroxyl in position 3 [49].”
In the line 269, the following text was added:
“The group of the flavanones have the C ring saturated and also are named dihydroflavones because the only difference with the flavones is that the double bond between positions 2 and 3 is saturated [49].”
In the line 309, the following text was added:
“The flavan-3-ols are also named dihydroflavonols, due always have in the position 3 of the C ring a hydroxyl group [49]. On the other hand, isoflavones also knowing as phytoestrogens [87].”
- In the introduction the impression is given that the review is about nutraceuticals. But most falvones have a very bad bioactivity and uptake. This is also shown by most experiments on mouse models described in this review, in which the flavanoids are topically applicated. Therefore I would suggest to make a chapter of its own, in which the best way of application is described. A good start is the section: line 340-370. It is compeletely unfitting at the current position (because it is not really linked to isoflavones), but already contains most statements
Reply:
Regarding “In the introduction the impression is given that the review is about nutraceuticals.”
To specify the focus of the introduction and avoid the impression that the review is about nutraceuticals, in line 61, the following text was added:
“…the primary strategy for prevention of photoaging is photoprotection, and the secondary treatment is by the use of exogenous antioxidants and other compounds that cannot be synthesized in our body [7].”
In line 81, the following paragraph was added:
“natural resinous product that bees make using material obtained from multiple botanical sources; it is mixed with beeswax and enzymes secreted by bees through their salivary glands [14]. Propolis is a sticky, flexible and soft resin at warm temperatures, or brittle and hard at cold temperatures, it also presents various colors such as red, green or brown [15, 16]. Generally, the chemical composition of propolis is 50% resin, 30% wax, 10% essential oils, 5% pollen and 5% other substances [17]. This natural product has been reported to present approximately 300 different compounds [18]. The characteristic chemical groups characterized in propolis are flavonoids, steroids, phenolic acids or their esters, terpenes, stilbenes, aromatic aldehydes and alcohols as well as fatty acids [18, 19]. It is also known that both the chemical composition and the biomedical effect of propolis have a very high variability according to the collection region, the sources of vegetation in the area and the seasons [20, 21].”
In the line 83, the following text was added:
“Of the different components that make up propolis, flavonoids stand out for having a great anti-inflammatory and antioxidant activity, among others. Furthermore, the plant sources of the flavonoids are highly variable since they depend directly on the geographic region and the flora that bees visit, in fact the presence and abundance of flavonoids is highly variable in propolis from different countries [20, 21].”
In the line 84, the following paragraph was added:
The flavonoids have a well-known chemical structure which consists of 15 carbon atoms that are arranged to form three aromatic rings named A, B, C. The B-ring is linked to the A-ring by a three carbon bridge that binds with one oxygen and two carbons of the A-ring thus forming the C-ring (Figure 1). Based on the different functional groups and level of oxidation in the C-ring, and different connections between B-ring and C-ring, flavonoids are classified into several groups such as flavones, flavonols, flavan-3-ols, isoflavones, flavanones, anthocyanidins, chalcones, and aurones [34].”
In the line 87 added "Figure 1. Basic flavonoid structure."
Regarding “But most falvones have a very bad bioactivity and uptake. This is also shown by most experiments on mouse models described in this review, in which the flavanoids are topically applicated. Therefore I would suggest to make a chapter of its own, in which the best way of application is described. A good start is the section: line 340-370. It is compeletely unfitting at the current position (because it is not really linked to isoflavones), but already contains most statements”
We agree with you, there is still a great abyss in terms of the best form of administration or application of the different flavonoids to achieve the expected effect, this also depends on the place of action of the required flavonoid activity, either at a systemic or local level. Therefore, the various ways in which flavonoids can exert their best activities are still under study, whether encapsulated to reach the intestine through oral administration or combined with different vehicles to offer the best absorption and bioavailability at the level of the skin.
Therefore, in line 350 of the text, the following paragraph was added:
“In this regard, it is known that some types of flavonoids administered or applied directly do not show good biological activity, however, an effort has been made to improve the biological activity of flavonoids by investigating different forms of conjugation or com-bination with different substances such as sepharose beads, nanoparticles or gels based on microemulsions, sodium alginate or poly (vinyl) alcohol or even liposomes as amphipathic transporters of bioactive compounds, such as for example EGCG, quercetin, naringenin, among others [74, 88, 99-101]. An example of this is the research conducted by Esposito et al. in 2020 [99], who has been studied a novel series of hybrid hydrogels at different ratios based on sodium alginate and poly(vinyl) alcohol quercetin-loaded. The permeability of quercetin through the skin showed different penetration/permeation profiles according to the hydrogel’s blend, which will allow the selection of hydrogels ratio for a best local and prolonged skin effect, making the use of these hydrogels a promising solution for the delivery of flavonoids for the treatment of skin ageing and inflammation. Another example is the study carried out in 2020 by Parashar et al. [100], who mention that the topical delivery system for of sericin gel loaded with microemulsion that containing naringenin, displayed higher retention and deposition of naringenin in the skin. Moreover, this is a propitious topical delivery system for naringenin for pre-venting or inhibiting UV-induced skin aging, that displayed enhanced therapeutic potential when compared with plain naringenin. Other research mentions that liposomes are nanocarriers that are used to incorporate bioactive compounds or drugs to treat some skin diseases. Liposomes are membranes of different sizes (from micro to nanometers) mainly composed of cholesterol and phospholipids, forming structures similar to cell membranes. Also, they are composed of a lipid bilayer and an aqueous nucleus that gives them their amphiphilic property. Furthermore, to improve permeability in the skin, edge activators are added to the liposomes, which reduces the stiffness of the bilayer structure making it deformable. However, more research is needed regarding the efficacy of loaded liposomes to transport different bioactive compounds to study their release through the skin [101]. It should be noted that it is still a challenge to find and standardize the best method of administration or application of flavonoids in a safe and efficient way to achieve their best expected biological activity for clinical application.”
In the line 258 the following paragraph was added:
“Quercetin is a very abundant flavonol in plant sources, making it one of the most studied flavonoids and for which there is sufficient evidence in vitro and in vivo studies of its beneficial effects on health. The effects of quercetin and its derivatives on photoaging should be considered as a viable and interesting alternative since they have effects at different levels of the pathological processes of photoaging. However, one of its main limitations is to find a vehicle or administration route that ensures its correct absorption since it is very limited. In this sense, various strategies have been developed, of which the design of nanoparticles with quercetin stands out, which apparently improve their assimilation [78-80]. Therefore, it is necessary to carry out more studies that can demonstrate whether nanoparticles with this flavonol improve the effects on ROS, activation of signaling pathways (NF-κB, Akt, JNK, ERK and AP-1), and the production of pro-inflammatory cytokines. At the same time, it is urgent that well-controlled clinical studies be carried out in order to find an adequate dose that guarantees the desired pharmacological effects without adverse effects.”
Furthermore, we consider that the objective of this review is not focused on describing which is the best form of application, absorption or bioavailability of the different flavonoids, since all this information would be enough to publish a review on these topics, as can be found in these recent papers:
- Castañeda-Reyes, E. D., de Jesús Perea-Flores, M., Davila-Ortiz, G., Lee, Y., & de Mejia, E. G. (2020). Development, characterization and use of liposomes as amphipathic transporters of bioactive compounds for melanoma treatment and reduction of skin inflammation: A review. International journal of nanomedicine, 15, 7627.
- Propolis should be explained in more detail. It should be stated that it is an undefined mixture of flavaonouds that differ between bee hives and therefore regions.
Reply:
In the line 81 the following paragraph was added:
“natural resinous product that bees make using material obtained from multiple botanical sources; it is mixed with beeswax and enzymes secreted by bees through their salivary glands [14]. Propolis is a sticky, flexible and soft resin at warm temperatures, or brittle and hard at cold temperatures, it also presents various colors such as red, green or brown [15, 16]. Generally, the chemical composition of propolis is 50% resin, 30% wax, 10% essential oils, 5% pollen and 5% other substances [17]. This natural product has been reported to present approximately 300 different compounds [18]. The characteristic chemical groups characterized in propolis are flavonoids, steroids, phenolic acids or their esters, terpenes, stilbenes, aromatic aldehydes and alcohols as well as fatty acids [18, 19]. It is also known that both the chemical composition and the biomedical effect of propolis have a very high variability according to the collection region, the sources of vegetation in the area and the seasons [20, 21].”
- It would be interesting to read what are the main plant sources of the flavanoids presented in this review
Reply:
We consider that adding more information on this variable could distract the reader from the main objective, since we would have to go deeply into the type of vegetation in the different geographical regions of the propolis in which the flavonoids mentioned in the review have been reported.
However, trying to clarify this point, in the line 82, the following text was added:
“Of the different components that make up propolis, flavonoids stand out for having a great anti-inflammatory and antioxidant activity, among others. Furthermore, the plant sources of the flavonoids are highly variable since they depend directly on the geographic region and the flora that bees visit, in fact the presence and abundance of flavonoids is highly variable in propolis from different countries [20, 21].”
- Would it be possible to somehow rankt the flavanoids according their effect on photoaging and psoriasis?? I know that this is difficult…….but would be interesting
Reply:
We include this answer as part of the paragraph that we added in line 761:
“If we analyzed the flavonoids that have been tested both photoaging and psoriasis found that some of them share properties that could be used in both pathologies; in the different subgroups of flavonoids, we found compounds that have been tested in parallel studies according to know their anti-photoaging and anti-psoriatic properties. Flavones and flavonols are the subgroups that present the majority of compounds tested in both illnesses (Tables 1-2, and 6-7); moreover, both types of flavonoids also share some compounds tested in both pathologies such as the flavones luteolin [39, 51, 53-55, 120-122], and chrysin [35, 58] and the flavonols kaempferol [70, 117], quercetin [70, 74, 76, 115, 122, 126], and fisetin [81, 127]. Although in the different studies in which these flavonoids have been evaluated the aims are different, some in vitro models used are similar between them; and it is in this point that we found the anti-inflammatory properties on keratinocytes cell lines that display to reduce the levels and regulate the activation, and/or expression of different molecules involved in both photoaging and psoriasis [51, 53-55, 58, 120, 121, 127]; therefore, this mechanism should be taken in count, and to take advantage of the antioxidants properties of the flavonoids tested on photoaging models [38, 50, 53, 55, 58, 81] to be applied in psoriasis models. Finally, the flavanones hesperidin [83, 84, 129], and naringenin [8, 85, 86, 128]; the flavan-3-ol EGCG [88-90, 130, 131]; and the isoflavone genistein [97, 132], also are tested in both illnesses; although it should be mentioned that the number of flavonoids studied of these subgroups is lower in comparison with the flavones and flavonols; therefore, an ranking to evaluate their anti-photoaging, and anti-psoriatic properties it becomes difficult to establish because there are so many signaling pathways that are not taken into account on in vitro and in vivo experimental models of both pathologies.”
Minor points:
- Line 63: Most skin diesaes have nothing to do with solar radiation
Reply:
In line 64 the text was corrected, and the following was added:
“…many external agents have the ability to cause skin diseases, some of these conditions have a diverse etiology…”
- Line 92: By definition: Photaging is the result of chronic UVA and UVB exposure. In contrat to that skin aging is also cuased by different “agents”.
Reply:
In line 93 the text was corrected, and the following paragraph was added:
“In addition to chronological aging, some extrinsic factors as pollutants, cigarette smoke and exposure to sunlight can accelerate skin aging [40-42]. The epidermal thickening, deep wrinkles, dryness and loss of elasticity are features of photoaged skin [40-42]. Histologically, a photodamaged skin presents accumulation of amorphous elastic fibers and disorganized and fragmented collagen in the dermis [43, 44]. Photoaging is the result of chronic UV radiation exposure from the sun [6]. UV rays can be divided in UVC, UVB, and UVA according to the wavelengths of radiation [4]. UVC is absorbed in the stratosphere, and therefore, only UVA and UVB reach the surface of the earth (95% UVA and 5% UVB approximately) [6]. UVA irradiation triggers stimulation of matrix metal-loproteinase-1 (MMP-1) responsible for collagen degradation, characteristic of photoaged skin [45]. UVB can damage DNA directly through the formation of cyclobutane py-rimidine dimers (CPDs), this action is also observed by UVA rays but to a much lower extent. Furthermore, both UVA and UVB can damage DNA indirectly through the generation of reactive oxygen species (ROS) [46, 47].”
- Line 95: Oxidative reactions lead to the activation of MMPs, MMPs are no oxidative reaction.
Reply:
In line 96 the text was corrected, and the following was added:
“UVA irradiation triggers stimulation of matrix metalloproteinase-1 (MMP-1) responsible for collagen degradation, characteristic of photoaged skin [45]”
- Conclusions: Some pathways (e.g. Keap-1) are for the first time meantioned in the conclusion section. A conclusion should be a conclusion ;-)
Reply:
Keap-1 was corrected and discarded from the conclusions section. =-)

Reviewer 4 Report
On the manuscript: antioxidants-1484928 “Flavonoids present in propolis in the battle against photoaging 2 and psoriasis”
This review was focused on describing the properties and benefits of flavonoids from propolis on pathogenetic mechanisms involved in ageing and psoriasis. The information collected showed that the antioxidant and anti-inflammatory properties of flavonoids play a crucial role in the control and regulation of the cellular and biochemical alterations occurred in these diseases; moreover, flavones, flavonols, flavanones, flavan-3-ols, and isoflavones contained in different propolis samples are tested in both diseases. The authors concluded that the research carried out in the area of dermatology with bioactive compounds of different origins is of great relevance to developing preventive and therapeutic approaches.
Comments and suggestions:
- In Introduction section the authors who are not specialists in the field make statements that are not related to reality. For example, skin psoriasis does not cause pain but only discomfort. Psoriatic arthritis causes joint pain. Another example page 2 line 72-74 –this treatment can cause discomforts such as irritation, phototoxicity. What exactly does it mean?
- The whole article should be reorganized to match the title. First of all, I would detail the pathogenesis of the studied diseases and the mechanisms involved. Then, I would mention the role of different classes of flavonoids in these diseases and then I would detail the effect of flavonoids from propolis in the two conditions. I would have to specify exactly what propolis contains in the composition. The mention in detail of the effect of flavonoids from various plant sources in aging and psoriasis is not related to the propolis. Moreover, the mechanisms involved in ageing and the mechanisms are presented briefly, narratively. The effect of flavonoids should also be discussed in more detail in connection with these mechanisms.
- The authors should mention that the in vitro and in vivo models of psoriasis are imperfect and that these compounds can only interfere with certain aspects of the pathogenesis of the diseases. The pathogenesis of the diseases is much more complex, especially in the case of psoriasis.
- The authors should discuss the skin sensitizing potential of propolis that limits its use in clinical practice.
Author Response
Reviewer 4
This review was focused on describing the properties and benefits of flavonoids from propolis on pathogenetic mechanisms involved in ageing and psoriasis. The information collected showed that the antioxidant and anti-inflammatory properties of flavonoids play a crucial role in the control and regulation of the cellular and biochemical alterations occurred in these diseases; moreover, flavones, flavonols, flavanones, flavan-3-ols, and isoflavones contained in different propolis samples are tested in both diseases. The authors concluded that the research carried out in the area of dermatology with bioactive compounds of different origins is of great relevance to developing preventive and therapeutic approaches.
Comments and suggestions:
- In Introduction section the authors who are not specialists in the field make statements that are not related to reality. For example, skin psoriasis does not cause pain but only discomfort. Psoriatic arthritis causes joint pain. Another example page 2 line 72-74 –this treatment can cause discomforts such as irritation, phototoxicity. What exactly does it mean?
Reply:
Regarding “In Introduction section the authors who are not specialists in the field make statements that are not related to reality. For example, skin psoriasis does not cause pain but only discomfort. Psoriatic arthritis causes joint pain.”
In line 32 the text was corrected, and the following was added:
“…and deterioration of the lifestyle of people affected.”
Regarding “Another example page 2 line 72-74 –this treatment can cause discomforts such as irritation, phototoxicity. What exactly does it mean?”
In the line 73 the following paragraph was added:
“…there are diverse therapies for the treatment of moderate-to-severe psoriasis that includes phototherapy, systemic agents such as methotrexate, and cyclosporine; oral treatments as apremilast, and topical therapies. Nevertheless, although there are many treatments that can be effective and well-tolerated, patients affected by psoriasis often do not achieve a clearance of the skin affected by this disease, and therefore, they do not present symptom relief or improvements in their quality of life. The above, have negative effects in patients, so, when receiving treatments ineffective or poorly tolerated for a long time can lead to the development of a sustained underlying inflammation; as well as to the deterioration of skin signs and symptoms, and, on the other hand, patients can develop comorbidities related with psoriasis, such psoriatic arthritis, metabolic syndrome, obesity, diabetes, hypertension, cardiovascular disease, among others”
- The whole article should be reorganized to match the title. First of all, I would detail the pathogenesis of the studied diseases and the mechanisms involved. Then, I would mention the role of different classes of flavonoids in these diseases and then I would detail the effect of flavonoids from propolis in the two conditions. I would have to specify exactly what propolis contains in the composition. The mention in detail of the effect of flavonoids from various plant sources in aging and psoriasis is not related to the propolis. Moreover, the mechanisms involved in ageing and the mechanisms are presented briefly, narratively. The effect of flavonoids should also be discussed in more detail in connection with these mechanisms.
Reply:
Regarding “The whole article should be reorganized to match the title. First of all, I would detail the pathogenesis of the studied diseases and the mechanisms involved.”
In the line 93 the following paragraph was added:
“In addition to chronological aging, some extrinsic factors as pollutants, cigarette smoke and exposure to sunlight can accelerate skin aging [40-42]. The epidermal thickening, deep wrinkles, dryness and loss of elasticity are features of photoaged skin [40-42]. Histologically, a photodamaged skin presents accumulation of amorphous elastic fibers and disorganized and fragmented collagen in the dermis [43, 44]. Photoaging is the result of chronic UV radiation exposure from the sun [6]. UV rays can be divided in UVC, UVB, and UVA according to the wavelengths of radiation [4]. UVC is absorbed in the stratosphere, and therefore, only UVA and UVB reach the surface of the earth (95% UVA and 5% UVB approximately) [6]. UVA irradiation triggers stimulation of matrix metal-loproteinase-1 (MMP-1) responsible for collagen degradation, characteristic of photoaged skin [45]. UVB can damage DNA directly through the formation of cyclobutane py-rimidine dimers (CPDs), this action is also observed by UVA rays but to a much lower extent. Furthermore, both UVA and UVB can damage DNA indirectly through the generation of reactive oxygen species (ROS) [46, 47].”
In the line 438 the following paragraph was added:
“Currently, it is known that psoriasis is the result of a disturbance among innate and adaptative cutaneous immune responses. It has been shown that this autoimmune pathology is displayed on an inflammatory background in which the psoriatic plaque can develop in the dermal layer of the skin; therefore, there is an interaction among keratinocytes (which are the cell type that shapes the layer of the skin) with different cells involved in the innate and adaptative immune response. LL37, b-defensins, and S100 proteins are the most studied psoriasis-associated AMPs (antimicrobial peptides), of which, LL37 also named cathelicidin, has been associated with a pathogenic role in psoriasis; this stimulates TLR-9 in plasmacytoid dendritic cells (pDCs); and starting the development of the psoriatic plaque in which there is a stimulation to produce of type I IFN (IFN-a and IFN-b), which, in turn, promotes the phenotypic maturation of myeloid dendritic cells (mDCs), that are implicated in Th1 and Th17 differentiation and function, as well as the production of IFN-g and IL-17; on the other hand, when activated mDCs migrate into draining lymph nodes, it secrets TNF-a, IL-23, and IL-12; in the case of IL-23 and IL-12 it is knowing that can modulate the differentiation and proliferation of Th17 and Th1 respectively. The maintenance phase of psoriatic inflammation is driven by different T cell subsets in which Th17 cytokines (IL-17, IL-21, and IL-22) activate the proliferation of keratinocytes in the epidermis; moreover, the proliferation of these cells also are stimulated by TNF-a, IL-17, and IFN-g; and therefore, keratinocytes are directly implicated in the inflammatory cascade by means of the secretion of different cytokines like IL-1, IL-6, and TNF-a; chemokines, and AMPs [12, 118].”
Regarding “Then, I would mention the role of different classes of flavonoids in these diseases and then I would detail the effect of flavonoids from propolis in the two conditions.”
The purpose of the paper is not to relate the beneficial effects of flavonoids in both diseases since their etiology and pathology are different. Furthermore, if they were analyzed jointly, it would be difficult to emphasize the relevant or particular effects of flavonoids in each pathology.
However, in the line 761, the following paragraph was added:
“If we analyzed the flavonoids that have been tested both photoaging and psoriasis found that some of them share properties that could be used in both pathologies; in the different subgroups of flavonoids, we found compounds that have been tested in parallel studies according to know their anti-photoaging and anti-psoriatic properties. Flavones and flavonols are the subgroups that present the majority of compounds tested in both illnesses (Tables 1-2, and 6-7); moreover, both types of flavonoids also share some compounds tested in both pathologies such as the flavones luteolin [39, 51, 53-55, 120-122], and chrysin [35, 58] and the flavonols kaempferol [70, 117], quercetin [70, 74, 76, 115, 122, 126], and fisetin [81, 127]. Although in the different studies in which these flavonoids have been evaluated the aims are different, some in vitro models used are similar between them; and it is in this point that we found the anti-inflammatory properties on keratinocytes cell lines that display to reduce the levels and regulate the activation, and/or expression of different molecules involved in both photoaging and psoriasis [51, 53-55, 58, 120, 121, 127]; therefore, this mechanism should be taken in count, and to take advantage of the antioxidants properties of the flavonoids tested on photoaging models [38, 50, 53, 55, 58, 81] to be applied in psoriasis models. Finally, the flavanones hesperidin [83, 84, 129], and naringenin [8, 85, 86, 128]; the flavan-3-ol EGCG [88-90, 130, 131]; and the isoflavone genistein [97, 132], also are tested in both illnesses; although it should be mentioned that the number of flavonoids studied of these subgroups is lower in comparison with the flavones and flavonols; therefore, an ranking to evaluate their anti-photoaging, and anti-psoriatic properties it becomes difficult to establish because there are so many signaling pathways that are not taken into account on in vitro and in vivo experimental models of both pathologies.”
Regarding “I would have to specify exactly what propolis contains in the composition.”
In the line 81, the following paragraph was added:
“natural resinous product that bees make using material obtained from multiple botanical sources; it is mixed with beeswax and enzymes secreted by bees through their salivary glands [14]. Propolis is a sticky, flexible and soft resin at warm temperatures, or brittle and hard at cold temperatures, it also presents various colors such as red, green or brown [15, 16]. Generally, the chemical composition of propolis is 50% resin, 30% wax, 10% essential oils, 5% pollen and 5% other substances [17]. This natural product has been reported to present approximately 300 different compounds [18]. The characteristic chemical groups characterized in propolis are flavonoids, steroids, phenolic acids or their esters, terpenes, stilbenes, aromatic aldehydes and alcohols as well as fatty acids [18, 19]. It is also known that both the chemical composition and the biomedical effect of propolis have a very high variability according to the collection region, the sources of vegetation in the area and the seasons [20, 21].”
We decided to use a brief definition because we do not intend to delve too deeply about propolis but rather to use it as a starting point since we intend to focus the review on the flavonoids that have been reported in propolis. And the other reason is to avoid duplication, since we have defined it in other of our publications on propolis, as mentioned below:
- Rivera-Yañez, N .; Rivera-Yañez, C.R .; Pozo-Molina, G .; Méndez-Catalá, C.F .; Méndez-Cruz, A.R .; Nieto-Yañez, O. Biomedical properties of propolis on diverse chronic diseases and its potential applications and health benefits. Nutrients 2021, 13, 78.
- Rivera-Yañez, N .; Rivera-Yañez, C.R .; Pozo-Molina, G .; Méndez-Catalá, C.F .; Reyes-Reali, J .; Mendoza-Ramos, M.I .; Méndez-Cruz, A.R .; Nieto-Yañez, O. Effects of propolis on infectious diseases of medical relevance. Biology 2021, 10, 428.
- Rivera-Yañez, N., Rodriguez-Canales, M., Nieto-Yañez, O., Jimenez-Estrada, M., Ibarra-Barajas, M., Canales-Martinez, MM, & Rodriguez-Monroy, MA (2018). Hypoglycaemic and antioxidant effects of propolis of Chihuahua in a model of experimental diabetes. Evidence-Based Complementary and Alternative Medicine, 2018.
Regarding “The mention in detail of the effect of flavonoids from various plant sources in aging and psoriasis is not related to the propolis.”
We avoid emphasizing this relationship, as it could confuse the reader and it would seem that we are mainly focused on propolis and we really only want to focus on the flavonoids that have been found in this. In addition, from line 414 at 428 we have a section where we review the effect of propolis on photoaging. Also, we add the following paragraph on line 418 to complement this information:
““Besides, it is important to highlight that there is very little information regarding the safety of propolis, since it is a subject of study little investigated, it was found that propolis is considered safe by Diniz et al. [114], based on the results obtained in the biochemical safety profile in your research. However, it is clear that more studies is needed on this subject, as well as on many of the natural products used as complementary treatments in various diseases. In addition, it is important to mention that it should start with the implementation of more investigations to establish and propose methods of analysis and extraction of the different compounds present propolis around the world, such as flavonoids, to be able to compare in a more efficient way the biomedical activities of the propolis and its bioactive components [23]. Furthermore, the little research on the safety and standardization of propolis are some of the limitations for its use in clinical practice.”
Furthermore, we do not mention propolis in psoriasis since there is only one study specifically on propolis and psoriasis, but this is from more than 23 years ago and it would be difficult to make a section at review with so little information. This paper is mentioned below:
- Ledón, N., Casacó, A., González, R., Merino, N., González, A., & Tolón, Z. (1997). Antipsoriatic, anti-inflammatory, and analgesic effects of an extract of red propolis. Zhongguo yao li xue bao= Acta Pharmacologica Sinica, 18(3), 274-276. PMID: 10072950.
However, in the line 82, the following text was added:
“Of the different components that make up propolis, flavonoids stand out for having a great anti-inflammatory and antioxidant activity, among others. Furthermore, the plant sources of the flavonoids are highly variable since they depend directly on the geographic region and the flora that bees visit, in fact the presence and abundance of flavonoids is highly variable in propolis from different countries [20, 21].”
Regarding “Moreover, the mechanisms involved in ageing and the mechanisms are presented briefly, narratively. The effect of flavonoids should also be discussed in more detail in connection with these mechanisms.”
In the line 125 the following paragraph was added:
“Particularly, the methodology used in this study is very interesting since they use human skin explants, which is a good study model to largely recreate various aspects of photoaging pathology, however, the authors only evaluated MMP- 1 and IL-6, when they could have gotten much more information on other parameters. Oliveira et al propose a model from skin explants obtained from healthy donors who underwent otoplasty surgery, where they performed a histological evaluation and measured ROS, MDA, MMP-1, MMP-8, MPO as well as macrophage in the tissue; the evaluation of all these parameters allows a better explanation of the mechanisms regulated by different treatments. They even used levels of UVA exposure compatible with a summer in Brazil, avoiding the models of excessive UVA radiation, which is not compatible with our daily sun exposure [52]. Ex vivo studies allow to recreate the mechanisms of photoaging to a great extent, however, these have their benefits and limitations since no in vitro, in vivo or ex vivo model is completely faithful to the pathological and clinical conditions of human diseases. To eliminate these limitations, it will be necessary to conduct well-controlled clinical studies.”
In the line 145 the following paragraph was added:
Luteolin is capable of regulating some important pathological mechanisms of photo aging such as the generation of ROS and MMP-1, this is important to continue studying this flavone and to be able to establish it as a complementary treatment in the future, because, these activities are also present in some retinoids such as vitamin A, E and safranal, among others; which are currently one of the main candidates in the treatment against photoaging [56, 57].”
In the line 155 the following paragraph was added:
“Although the effects demonstrated by chrysin are positive and interesting, there is still little information regarding its anti-photoaging effects, so it is necessary to carry out more in vitro and in vivo studies to evaluate the different alterations related to photoaging that justify proposing chrysin as a serious alternative against this condition.”
In the line 181 the following paragraph was added:
“It is notable that baicalin has a great antioxidant effect and that it is capable of regulating the levels of oxidative stress and other various parameters in in vitro and in vivo models, which is very favorable in the search for alternatives that complement the therapy against photoaging. Since today the use of sunscreens, protective clothing, avoiding sun's harmful radiation thereby reducing the progression of skin aging. However, it is precisely molecules with antioxidant properties that help to develop resistance to oxidative stress and slows down the process of skin aging; therefore, baicalein and flavonoids with antioxidant properties are a serious proposition in photoaging therapy [63]”
In the line 193 the following paragraph was added:
“It can be see that the administration of this flavonoid acts on cytokines and transcription factors that are directly related to alterations caused by exposure to UVA and UVB rays, but the most relevant aspect is that they suggest that baicalein regulates these alterations by signaling the TLR4, which opens the door to carry out studies in combination with other flavonoids such as quercetin, resveratrol, apigenin and luteolin [66-69] in order to find a synergistic effect for a better regulation of this signaling pathway in photoaging.”
In the line 218 the following paragraph was added:
“The results reported for galangin and kaempferol are good and favorable, however they are still very partial and not very integrative to be considered a real alternative against photoaging. Therefore, it is still necessary to carry out more studies focused on evaluating more characteristics of this disease, such as the levels of ROS, MPO, pro-inflammatory cytokines and even the use of models that allow a histological analysis of the tissue [73]. The challenges for galangin, kaempferol and myricetin are still several, but the future investigations will help conclude if these flavonoids can be used against photoaging.”
In the line 258 the following paragraph was added:
“Quercetin is a very abundant flavonol in plant sources, making it one of the most studied flavonoids and for which there is sufficient evidence in vitro and in vivo studies of its beneficial effects on health. The effects of quercetin and its derivatives on photoaging should be considered as a viable and interesting alternative since they have effects at different levels of the pathological processes of photoaging. However, one of its main limitations is to find a vehicle or administration route that ensures its correct absorption since it is very limited. In this sense, various strategies have been developed, of which the design of nanoparticles with quercetin stands out, which apparently improve their assimilation [78-80]. Therefore, it is necessary to carry out more studies that can demonstrate whether nanoparticles with this flavonol improve the effects on ROS, activation of signaling pathways (NF-κB, Akt, JNK, ERK and AP-1), and the production of pro-inflammatory cytokines. At the same time, it is urgent that well-controlled clinical studies be carried out in order to find an adequate dose that guarantees the desired pharmacological effects without adverse effects.”
In the line 292 the following paragraph was added:
“Hesperidin and hesperetin have similar effects since they decrease ROS levels, collagen degradation and MMP activity, however in these studies no histological or ex vivo models are used, which would help to have more certainty about the benefits of these flavanones in photoaging. On the other hand, it would be interesting to determine if the co-administration of hesperidin and hesperetin increases their activity and thus achieve better protection against damage caused by UV radiation.”
In the line 328 the following paragraph was added:
“The properties of EGCG, like the various flavonoids mentioned in this work, have properties that attend to some of the molecular, biochemical and cellular disorders caused by photoaging. Which is very interesting since several of the treatments developed with the damage generated by UV radiation have a cosmetic approach instead of a therapeutic one, that is, they are designed to reduce wrinkles and spots on the skin, leaving aside the severe damage at cellular level caused by radiation and ROS, which can cause other diseases such as skin cancer [92-95]. Therefore, flavonoids could be proposed in the design of new cosmeceutical products, which have a cosmetic approach as well as a therapeutic one, that is, they attend to health and beauty. These products are relatively new to the dermatology industry so there are few alternatives available on the market.”
In the line 577 the following paragraph was added:
“Of general form all flavones reported with effects related to the anti-psoriatic activity were tested on keratinocytes cell lines in which most of the flavones [35, 120-123, 125] inhibit the cell growth of these cells; this property that displays the compounds tested it is through distinct mechanism that involved the inhibition of different factors of transcription or arrest of the cellular cell cycle. Moreover, in vivo evaluation of flavones decreases the psoriatic area in both histological and PASI measurements (Table 6). In this line, it becomes clear that the reduction or inhibition of key cytokines in the pathogenic mechanism of psoriasis (NF-kB, TNF-a, IL-1b, IL-23, and IL-17) plays an important role in the activity of the flavones. Although each compound presents different forms and has different activity on chemokines, cytokines, transcription factors, and molecules related to the development of psoriasis.”
In the line 651 the following paragraph was added:
“To difference with the flavones, the studies related to the flavonols present different ways to evaluate their anti-psoriatic effects (Table 7). Both quercetin and kaempferol reduce the psoriatic symptoms in the murine imiquimod-induce skin damage like psoriasis model, and although both flavonoids share the decrease of TNF-a, IL-6, and IL-17 (cytokines related with psoriasis) the mechanisms involved in the activity of both compounds it is different because quercetin is related with their antioxidant properties and their capacity to regulates the expression of key molecules in psoriasis, and on the other hand, the anti-psoriatic activity of kaempferol is closely related with the modulation of inflammation in the psoriatic tissue. Finally, only fisetin did not test on in vivo experimental model; nevertheless, displays different properties related to the inhibition of the cellular growth, signaling, and inflammatory profile of different cell lines used to test the mechanism of action that lead to the development of psoriasis, and the properties of fisetin could be used to determine their possible anti-psoriatic activity on in vivo experimental models.”
In the line 733 the following paragraph was added:
“Although the compounds tested belong to different subgroups of flavonoids, all of them have anti-psoriatic properties both in vitro and in vivo. It is interesting that too despite the differences among they all compounds have the capacity to reduce the psoriatic symptoms to histological level and have in common that the possible mechanisms that present are related to the regulation of inflammatory process in the damaged tissue by psoriasis. It is important to denote that only the anti-psoriatic activity of EGCG is related with their antioxidant activities [131], unlike the flavanones as naringenin, and hesperidin, or the isoflavone genistein that also have strong antioxidant properties [49, 87]. Therefore, the research of the antioxidant effects of these subgroups of flavonoids should be explored in future works to have a better under-standing of the anti-psoriatic activities of these flavonoids both in vivo and in vitro experimental models.”
- The authors should mention that the in vitro and in vivo models of psoriasis are imperfect and that these compounds can only interfere with certain aspects of the pathogenesis of the diseases. The pathogenesis of the diseases is much more complex, especially in the case of psoriasis.
Reply:
In the line 438 the following paragraph was added:
“Different experimental models to study psoriasis and the related mechanisms to this autoimmune disease have been developed both in vitro and in vivo; these, are used by many researchers to investigate the pathogenic mechanisms of psoriasis. Although it is worth mentioning that both experiment models have limitations such as the absence of blood vessels and microenvironment on in vitro models, and the differences of thickness, epidermis architecture, and immunological mechanisms involved on murine in vivo models with respect to the pathogenesis present in the human dermis. Nevertheless, both types of experimental models have been helpful to test the effectiveness and the development of new therapeutic agents [119].”
- The authors should discuss the skin sensitizing potential of propolis that limits its use in clinical practice.
Reply:
As we mentioned, we do not intend to delve too deeply about propolis.
However, in the line 418 the following paragraph was added:
“Besides, it is important to highlight that there is very little information regarding the safety of propolis, since it is a subject of study little investigated, it was found that propolis is considered safe by Diniz et al. [114], based on the results obtained in the biochemical safety profile in your research. However, it is clear that more studies is needed on this subject, as well as on many of the natural products used as complementary treatments in various diseases. In addition, it is important to mention that it should start with the implementation of more investigations to establish and propose methods of analysis and extraction of the different compounds present propolis around the world, such as flavonoids, to be able to compare in a more efficient way the biomedical activities of the propolis and its bioactive components [23]. Furthermore, the little research on the safety and standardization of propolis are some of the limitations for its use in clinical practice.”

Round 2
Reviewer 4 Report
Comments and suggestions: The authors have satisfactory answered to all questions mentioned.